# CAN TRANSFORMERS PREDICT SYSTEM COLLAPSE IN DYNAMICAL SYSTEMS?

## ABSTRACT

Transformer architectures have recently surged as promising solutions for non-linear dynamical systems, proposed as foundation models capable of zero-shot dynamics reconstruction and forecasting. Despite this success, it remains unclear whether they can truly serve as reliable digital twins of dynamical systems, i.e., whether they capture the underlying physics in distinct parameter regimes, especially in parameter regimes from which no training data is taken. In nonlinear dynamics, reservoir computing (RC) has already demonstrated broad success, especially in parameter-space extrapolation, as it is intrinsically a dynamical system capable of capturing not only the dynamical climate of the target system but more importantly, how the climate changes with parameter. Transformers, in contrast, rely on permutation-invariant attention mechanisms that can limit their ability to capture how temporal structure changes with parameter. To determine if transformers have the capability of dynamics extrapolation, we take predicting catastrophic collapse, which occurs when a bifurcation parameter crosses a critical threshold, as a benchmark task. Models are trained on trajectories in normal parameter regimes and then tested on parameters in an unseen regime with system collapse. Our results show that transformers, across configurations, consistently fail to capture collapse, while RC reliably predicts the transitions. This surprising finding raises questions about the generalization ability of transformers to dynamical systems, a topic warranting future research.

## 1 INTRODUCTION

In nonlinear dynamical systems, catastrophic collapse is determined by underlying bifurcation parameters. In real-world scenarios, these parameters often vary over time. For example, for a potential collapse of the Atlantic Meridional Overturning Circulation, while the dynamical mechanism is a simple saddle-node bifurcation, a shift in hidden parameters can induce a state flip that disrupts the warm climate of Europe (Zhai et al., 2024b). In electrical power systems, voltage collapse can lead to large-scale blackouts, driven by transient chaos near bifurcations (Dhamala & Lai, 1999). A particularly important case is the crisis bifurcation, where the destruction of a basin boundary leads to the collapse of a chaotic attractor into a chaotic transient (Grebogi et al., 1983; Lai & Tél, 2011).

Often, accurate governing equations are rarely available in practice, while time-series data are ubiquitous. In light of this, machine learning-based digital twins are increasingly used as surrogates for target dynamical systems, enabling simulations of parameter changes, control, and inverse modeling (Chakraborty & Adhikari, 2021). Recent empirical studies Kong et al. (2021a); Patel et al. (2021); Kong et al. (2023); Panahi et al. (2024); Yan et al. (2024); Zhuge et al. (2025) have demonstrated that digital twins can predict critical transitions (or tipping points as another terminology), one of the most significant challenges in nonlinear dynamics. These approaches typically involve training models on system trajectories under safe parameter regimes, and then testing on trajectories generated under unseen parameters with critical transition and tipping. A parameter channel is often injected to provide physical context and improve the performance. Among these, reservoir computing (RC) has been especially successful: as a dynamical system in itself, it can naturally imitate the target system and adapt to out-of-domain regimes. A potential perspective for interpreting RCs is that their dynamics are generally an embedding of the target dynamical system, which can be deemed as generalized synchronization (Hart et al., 2020; Gauthier et al., 2021). However, RC also suffers from limitations: the finite network size constrains its predictive capability.

The past few years have witnessed the remarkable success of the transformer architecture, the most influential sequence model to date (Vaswani et al., 2017). Initially dominating natural language processing, transformers have since percolated into fields such as time-series forecasting (Su et al., 2025), video analysis (Selva et al., 2023), and increasingly, nonlinear dynamics Zhai et al. (2025); Zhang & Gilpin (2024); Lai et al. (2025). One key advantage of transformers is their generalization ability, particularly in out-of-domain tasks, as seen in phenomena such as "grokking" in large language models (Power et al., 2022). In nonlinear dynamics, transformers have been applied to reconstruct unseen systems from sparse or random observations (Zhai et al., 2025), and time-series based large models have even been applied or built for zero-shot forecasting of new dynamics (Zhang & Gilpin, 2024; Lai et al., 2025). Yet, no theoretical guarantee exists that a trained transformer can perform as a faithful digital twin, either for the systems it was trained on or for unseen test systems. Indeed, a study Zhang & Gilpin (2025) suggests that even large pretrained models such as Chronos underperform simple baselines such as parroting. In mechanisms, this may not be surprising: RC resembles the target system by being a dynamical system itself, while the transformer relies on permutation-invariant self-attention, which is less sensitive to temporal ordering (Zeng et al., 2023). This raises a fundamental question: Can transformers serve as digital twins of dynamical systems?

To address this question, we focus on a classical yet significant challenge: machine-learning prediction of critical transition and tipping. Examples of collapse abound, from massive extinction in ecosystems and sudden lake pollution to power grid failures. In chaotic systems, short-term prediction is meaningful only up to several cycles of natural oscillations (or equivalently, several Lyapunov times), beyond which long-term trajectories diverge. In comparison, the ability to capture critical transitions under changing parameters becomes a decisive test of the ability of the model as a digital twin. Following prior works Kong et al. (2021a;b) on RC-based prediction of critical transitions, we replace RC with transformers and conduct a comprehensive evaluation.

Our results are surprising. Transformers train effectively on trajectories within safe parameter regimes, achieving strong multi-step prediction performance. However, when tested on unseen parameter values corresponding to collapse states, transformers fail to capture the transition, i.e., instead of predicting a state shift, they produce persistent oscillations as in the training regime. To study this further, we test four representative systems: a chaotic food chain system, a power system, the Ikeda map, and the Kuramoto-Sivashinsky equation, which vary in dimension and complexity. Despite extensive attempts to mitigate overfitting, such as broadening the training parameter regimes or adjusting the number of parameters in transformer, none of our transformer variants successfully predicted collapse in any system. These findings indicate that a critical rethink of the role of transformers in nonlinear dynamics may be necessary. Specifically, we make the following contributions:

1. To our knowledge, this is the first work to systematically challenge the effectiveness of transformers as digital twins in dynamical systems for anticipating system collapse.

2. We conduct comprehensive experiments across multiple systems and transformer settings, and benchmark against established solutions. While RC reliably succeeds at this task, standard transformer configurations consistently fail.

3. We propose critical transition and tipping prediction as a benchmark task for evaluating digital twins, complementing standard forecasting metrics, and highlight implications for future research.

To sum up, our study suggests that the widely reported generalization ability of transformers may be overstated, at least in the context of serving as digital twins for predicting critical transitions and tipping points. By contrast, RC remains highly effective, although its scalability limits prevent it from functioning as a general foundation model. Looking ahead, we hope this work stimulates renewed attention to physics-guided architectures, and inspires the development of next-generation machine learning frameworks that can faithfully capture not only the state evolution of dynamical systems, but also how their behavior shifts under parameter variation, thereby serving as reliable digital twins.

## 2 RELATED WORK

### 2.1 MACHINE LEARNING FOR DYNAMICAL SYSTEMS

Recurrent neural networks (RNNs) are designed to capture temporal dependencies in sequential data. However, Mikhaeil et al. (2022) demonstrate that standard RNNs are difficult to train on chaotic time series. Gated architectures such as LSTMs or GRUs (Vlachas et al., 2020), with strong stability constraints, can mitigate this issue by "locking" the network into stable dynamics. More recently, Hess et al. (2023) proposed a modified teacher forcing scheme for RNNs that strictly bounds gradients in chaotic regimes, enabling faithful attractor reconstruction. As another direction, researchers have found that RC provides high-performance, model-free prediction of chaotic systems at low cost (Pathak et al., 2018). Since its introduction, the success of RC in chaotic prediction has stimulated both theoretical and applied developments (Fan et al., 2021; Gauthier et al., 2021; Canaday et al., 2021; Flynn et al., 2023; Kim & Bassett, 2023; Zhai et al., 2023; Lin et al., 2024; Zhai et al., 2025). In parallel, transformers have been applied to dynamical systems, for example in reconstructing unseen dynamics from sparse observations (Zhai et al., 2025) and in large-scale zero-shot models for chaotic forecasting (Zhang & Gilpin, 2024; Liu et al., 2024; Lai et al., 2025). In addition, another related line of work is physics-informed neural networks (PINNs), which incorporate physical constraints into the network (Cuomo et al., 2022), and aligns with the idea of including system parameters as additional input channels to improve model performance. It is also worth noting that, Du et al. (2023) question the capability of transformers in optimal output estimation problem in dynamical systems.

### 2.2 MACHINE LEARNING FOR CRITICAL TRANSITIONS

Critical transitions (or tipping points) are abrupt and often irreversible changes in system state that occur once stability thresholds are crossed. Machine learning, in particular reservoir computing, has been introduced to this field, where the model is trained on trajectories from known parameters and then used to infer and predict critical transitions under unseen parameters (Kong et al., 2021a). Shortly thereafter, Kim et al. (2021) reported that reservoir computing can interpolate and extrapolate dynamics, effectively learning to infer global temporal structure from local examples. In parallel, Bury et al. (2021) demonstrated that deep learning can serve as an early-warning tool, outperforming traditional early-warning signal (EWS) methods. Since then, machine learning approaches have been studied systematically as digital twins of dynamical systems and predict critical transitions: models that evolve in parallel with the physical system, remain synchronized with observational data, and can be probed under different parameter regimes (Kong et al., 2021b; 2023). For example, Panahi et al. (2024) studied a data-driven framework for predicting tipping in nonautonomous systems, with real-world examples. Moreover, Huang et al. (2024) showed that deep learning models can predict rate-induced tipping. In addition, machine learning has been applied to reduce complex models and extract low-dimensional tipping mechanisms (Fabiani et al., 2024).

## 3 PARAMETER-AWARE MACHINE LEARNING FOR CRITICAL TRANSITION PREDICTION

In this section, we motivate and introduce parameter-aware machine learning methods for predicting critical transitions and tipping points. The overview of the framework is illustrated in Fig. 1. Panel (a) shows a schematic bifurcation diagram for a typical nonlinear system: as the bifurcation parameter varies, the qualitative behavior of the system changes accordingly. In the orange region (covered by the blue window), chaotic attractors and periodic oscillations coexist; we refer to this as the safe regime. At a critical parameter value (vertical black line), a catastrophic bifurcation (crisis) destroys the chaotic attractor, after which trajectories exhibit a short chaotic transient followed by collapse. In other words, to the left of the critical point $p_c$ the time series remains oscillatory (periodic or chaotic), whereas slightly to the right of $p_c$ the dynamics transition to collapse after a transient. Our main question is: given that the system is currently in a safe regime, can a data-driven model anticipate and predict collapse when the parameter drifts beyond the critical point?

We consider an extreme setting in which only three parameter values, $p_1, p_2, p_3$ (all in the safe regime), and their corresponding time series are available for training. The goal is to train a model

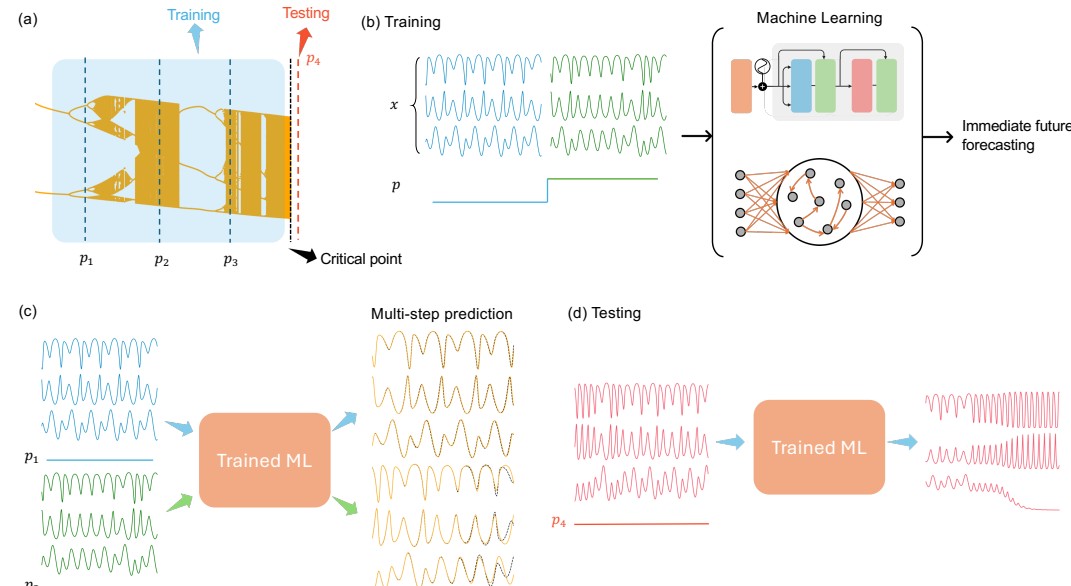

Figure 1: Parameter-aware machine learning framework for critical transition prediction. (a) Schematic bifurcation diagram. As the bifurcation parameter $p$ varies, the system remains in a safe regime (periodic/chaotic attractors) until a critical point $p_c$ (black vertical line), beyond which trajectories exhibit transient chaos followed by collapse. (b) Training. Models are trained on trajectories from safe parameters $p < p_c$, with the input formed by the time series and parameter channel. (c) Trained models are evaluated by multi-step predictions at training parameters. (d) Testing. Given a new unseen parameter $p_4 > p_c$, the model is tasked with forecasting forward. A successful prediction reproduces the collapse (transient followed by decay), rather than sustained oscillations.

that, when presented with a new parameter $p_4 > p_c$, correctly predicts future collapse. The training protocol is shown in Fig. 1(b) and follows a parameter-aware design: let $\mathbf{x}$ denote the system state time series and $\mathbf{p}$ the associated parameter channel; the model input is the concatenation $[\mathbf{x}; \mathbf{p}]$. During training we minimize a one-step-ahead prediction loss. However, this single-step accuracy is insufficient to certify learning in chaotic systems, thus we further validate the trained model via multi-step forecasting under the same training parameters, as shown in Fig. 1(c). A model is deemed well trained if it can produce accurate multi-step predictions over several oscillation cycles in the safe regime.

Finally, we assess the model on the critical transition task by providing the test parameter $p_4 > p_c$ together with the corresponding initial transient, and asking the model to forecast forward, as depicted in Fig. 1(d). A successful prediction is one in which the model correctly anticipates collapse (i.e., reproduces a transient followed by decay), rather than persisting in an oscillatory state. In what follows, we apply this evaluation protocol to three benchmark chaotic systems and compare two parameter-aware architectures: RC and transformer, under multiple neural network configurations. It is worth noting that, the generalization task here differs fundamentally from tasks in Natural Language Processing (NLP), as it requires sensitivity to parameter variation and stability thresholds, which may raise challenge for self-attention mechanisms.

## 4 METHODS

**Datasets.** We use chaotic systems simulated following Kong et al. (2021a) and Kong et al. (2021b) for fair comparison. The benchmark includes a three-dimensional food chain system, a four-dimensional voltage system, the discrete Ikeda map, and a sptio-temporal Kuramoto-Sivashinsky system. These systems do not satisfy sparsity conditions, and thus their governing equations cannot be identified by sparse regression methods (e.g., SINDy-type methods) (Wang et al., 2011; Brunton et al., 2016; Lai, 2021). Among them, the Ikeda map is the most difficult case, and yet no existing

method can faithfully recover its governing equations from data. More details about these systems can be found in ppendix A. By varying system parameters, each system can be driven from a stable to an unstable (collapsed) state.

**Models.** We adopt a parameter-aware machine learning framework based on both RC and transformers, where chaotic time series and the corresponding parameters are combined as model inputs. For RC, we follow the same simulation setup as Kong et al. (2021a) and Kong et al. (2021b), with network sizes typically smaller than 1000. A distinctive feature of RC is that the input weights and network adjacency remain fixed during training, and only the output weights are optimized. Under this scheme, the number of trainable parameters is on the order of $10^3$. For the transformer, we use a decoder-only autoregressive model. In the main experiments, we set the maximum input length $L_{\max} = 512$ and use order of $10^6$ trainable parameters. Detailed RC and transformer configurations are reported in the Appendix B.

**Pipelines.** To evaluate the ability of each model to predict critical transitions, we generate 500,000 data points for each bifurcation parameter, and for each system. For the continuous systems, one data point corresponds to roughly 1/50 of a cycle of oscillation. The task proceeds as follows: We first train the model under safe parameter regimes until it can achieve multi-step prediction performance comparable to the ground truth, and then we test the model on unseen parameter values expected to induce collapse. A successful prediction is defined as correctly forecasting the state shift (collapse). In our main experiments, for training of each system, we provide both the chaotic time series and three associated parameter values. The training dataset size differs substantially between RC and transformers: RC typically requires about $10^3 \sim 10^4$ points, whereas transformers require about $10^6$. In both cases, we use the minimal training size sufficient for accurate multi-step prediction in the training regime. To ensure statistical reliability, we repeat each experiment 50 times independently for each system. All training experiments were conducted on two servers, each equipped with 2 AMD EPYC 7763 CPUs and 12 NVIDIA RTX A6000 GPUs.

**Metrics.** We evaluate multi-step forecasting accuracy using the root mean square error (RMSE):

$$\text{RMSE}(X, \hat{X}) = \sqrt{\frac{1}{T_{\text{p}}d} \sum_{t=1}^{T_{\text{p}}} \sum_{j=1}^{d} \left(X_{t,j} - \hat{X}_{t,j}\right)^2}, \tag{1}$$

where $X \in \mathbb{R}^{T_{\text{p}} \times d}$ and $\hat{X} \in \mathbb{R}^{T_{\text{p}} \times d}$ denote the ground-truth and predicted trajectories, $T_{\text{p}}$ is the prediction horizon, and $d$ is the system dimension.

To assess performance in predicting critical transitions, we report the collapse prediction rate $P_c$. Within a search range of supercritical bifurcation parameters $p > p_c$, a trial is counted as "collapse" if the predicted trajectory falls below a predefined threshold $\theta$ and remains there for a sufficient duration. Formally,

$$P_c = \frac{1}{M} \sum_{m=1}^{M} \mathbf{1}\{\text{collapse detected in trial } m\}, \tag{2}$$

where $M$ is the number of independent trials (e.g., model seeds or warm-up segments).

## 5 RESULTS

To evaluate the effectiveness of transformer and RC in predicting critical transition and tipping, we conduct experiments on three benchmark chaotic systems: the food chain system, the voltage system, and the Ikeda map. For each system, trajectories at three bifurcation parameters within the chaotic or periodic regime are provided for training. The task is to predict dynamics at an unseen parameter value that induces collapse. A successful prediction is not the continuation of a periodic or chaotic attractor but rather a collapse, i.e., a sudden drop of the system state to a lower constant value. Additional experiments exploring transformer architectures and training settings can be found in Appendix F.

### 5.1 CRITICAL TRANSITION PREDICTION IN THE FOOD CHAIN SYSTEM

For clarity, we first present detailed results for the food chain system, with the other two systems deferred to the Appendix (See Appendix C). The chaotic food chain system provides a canonical

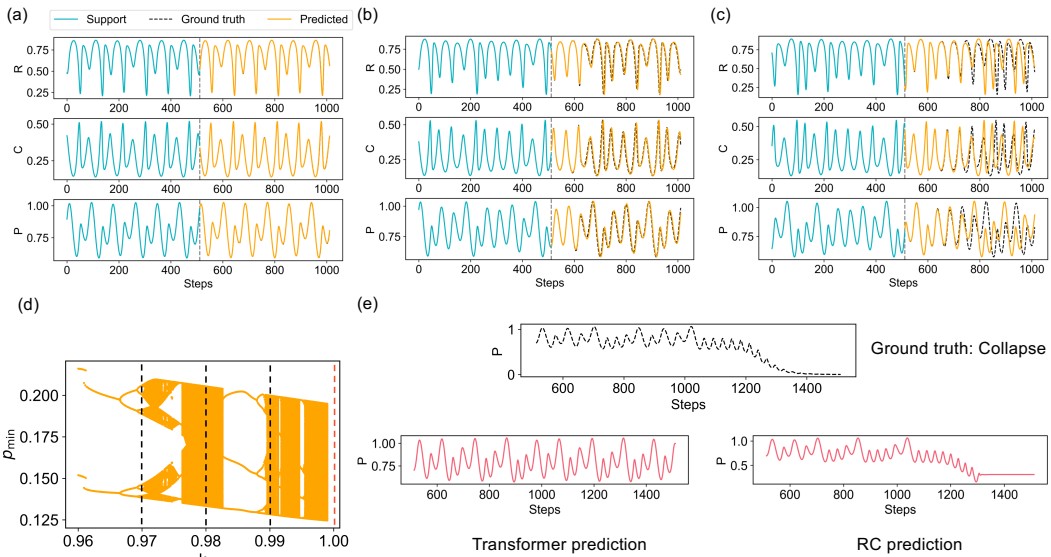

Figure 2: Critical transition prediction in the food chain system. (a-c) Multi-step predictions by the transformer for $K = 0.97, 0.98$, and $0.99$, respectively. Blue curves indicate warm-up input, and orange curves denote closed-loop transformer predictions. (d) Bifurcation diagram, with training parameters (black dashed) and the testing parameter beyond the critical point (red dashed). (e) Comparison of transformer and RC predictions at $K = 1.0 > K_c$. The ground truth trajectory collapses after a transient; RC reproduces this collapse, whereas the transformer continues oscillating.

ecological example where parameter drift can cause catastrophic collapse. It models the interactions among three species: resources $R$, consumers $C$, and predators $P$, governed by

$$
\begin{aligned}
\frac{dR}{dt} &= R\left(1 - \frac{R}{K}\right) - \frac{x_c y_c C R}{R + R_0}, \\
\frac{dC}{dt} &= x_c C\left(\frac{y_c R}{R + R_0} - 1\right) - \frac{x_p y_p P C}{C + C_0}, \\
\frac{dP}{dt} &= x_p P\left(\frac{y_p C}{C + C_0} - 1\right),
\end{aligned}
\tag{3}
$$

where $R, C, P$ are the population densities of the three species. The bifurcation parameter $K$ controls the environmental carrying capacity of the resource. As $K$ increases, the system undergoes a boundary crisis bifurcation, as illustrated in Fig. 2(d), with a critical point near $K_c \approx 0.99976$. For $K < K_c$, the system exhibits sustained chaos; for $K > K_c$, chaotic transients eventually collapse to extinction. Other constants $(x_c, y_c, x_p, y_p, R_0, C_0)$ are chosen to be ecologically reasonable and are remained fixed.

We train both models on chaotic trajectories at $K = [0.97, 0.98, 0.99]$, all within the safe regime, as illustrated in Fig. 2(d). During training, both RC and transformer achieve accurate multi-step forecasting over several Lyapunov times. The performance of transformer in this stage is shown in Fig. 2(a-c), where it closely tracks the true dynamics. Afterwards, we test the models at a parameter beyond the critical point ($K = 1.0 > K_c$). In the ground truth, the predator density $P$ undergoes transient chaos followed by collapse. Figure 2(e) shows the failure of the transformer, as it continues generating oscillatory trajectories resembling those in the training regime, and never predicting collapse. By contrast, RC initially follows the transient oscillations and subsequently forecasts the collapse, consistent with the ground truth. In this setting, none of our experiments with transformers resulted in success, despite extensive tuning of architectures and training ranges.

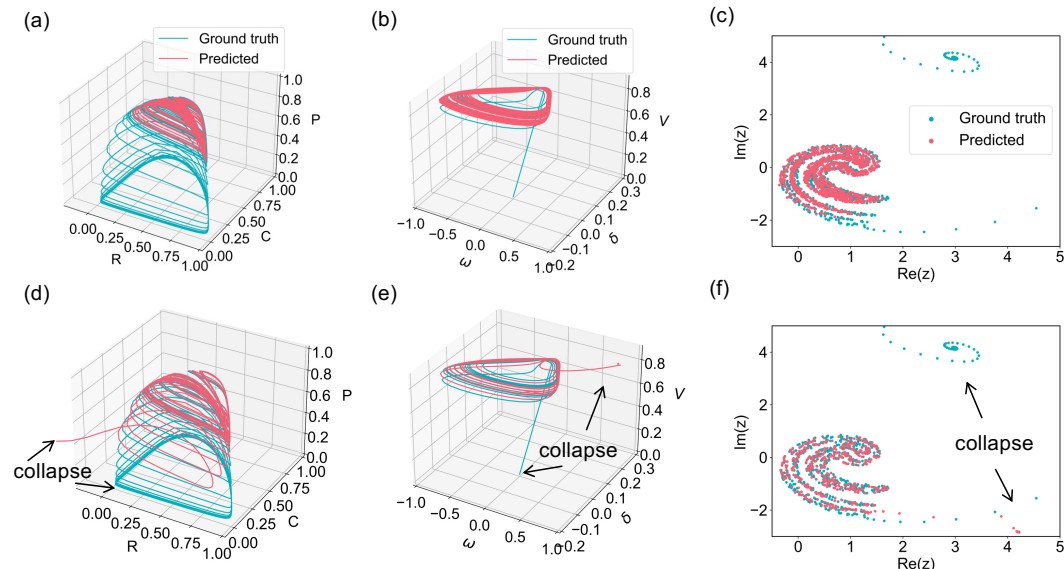

Figure 3: Long-term predictions across benchmark systems. Top and bottom rows are for transformer and RC predictions, respectively. (a, d) Food chain system. (b, e) Power system. (c, f) Ikeda map. Ground-truth dynamics undergo collapse beyond the critical bifurcation parameter, which is faithfully reproduced by RC but consistently missed by the transformer.

## 5.2 CRITICAL TRANSITION PREDICTION IN LONG-TERM ATTRACTORS

An important consideration is that the predictions in Fig. 2 are shown only over relatively short horizons. In practice, models may exhibit different reaction times to parameter changes. Although $K > K_c$, the deviation from the critical point may be small, and the model may not respond immediately. Indeed, we observed that across different trained RCs, and even across different warm-up segments for the same model, the predicted collapse occurred at distinct times under the same parameter. It is worth emphasizing that, to the best of our knowledge, no method can predict the exact timing or post-collapse state. At best, a well-trained model can indicate that a transition will occur within a future time window. This perspective is crucial for interpreting collapse prediction: one may ask whether the transformer's apparent "failure" in Fig. 2 might eventually collapse given a longer horizon, or conversely, whether RC's apparent success might eventually revert to oscillations.

To evaluate long-term behavior, we extend the predictions of both transformer and RC under the same experimental settings as before. For clarity, we illustrate results for all three benchmark chaotic systems in this section, while detailed system descriptions are provided in Appendix A. In brief, the power system is a four-dimensional electrical model in which chaotic fluctuations can cause voltage collapse as the load parameter $Q_1$ varies. The Ikeda map, in contrast, is a one-dimensional complex map describing the dynamics of a laser pulse in a nonlinear cavity, with the dimensionless input amplitude $\mu$ as its bifurcation parameter. For the power system, we train on three safe chaotic parameters $Q_1 = [2.98968, 2.98973, 2.98978]$ (Fig. 4d) and test at $Q_1 = 2.989830 > Q_{1c}$. For the Ikeda map, training is performed at $\mu = [0.91, 0.94, 0.97]$, and testing at $\mu = 1.01 > \mu_c$ (Fig. 5d).

Examples of long-term predictions are shown in Fig. 3. The first row presents transformer predictions, and the second row shows RC predictions, with the three columns corresponding to the food chain system, the power system, and the Ikeda map, respectively. For the power system, which has four variables, we visualize the last three; for the Ikeda map, which involves a single complex variable $z$, we plot its real and imaginary components. For fair comparison, both predictions of the two models are displayed on the same scale within each system. The results reveal a consistent pattern: the transformer persistently generates stable attractors and never transitions to collapse, even when the ground truth does. In contrast, RC successfully predicts collapse in all cases, as illustrated in panels (d–f). We note that in the power system, the ground truth collapse involves a sharp excursion

Table 1: Performance on critical transition prediction across systems

| system | model | RMSE↓ | $P_c$(collapse) ↑ | pred. crit. | gt crit. | $T_{train}$[s]↓ |
|---|---|---|---|---|---|---|
| Food chain | Transformer | 0.026 | 0.38 | 1.14958 | 0.99976 | 5744 |
| | Transformer$_b$ | 0.020 | 0.00 | N/A | 0.99976 | 5760 |
| | RC | 0.002 | 1.00 | 0.99986 | 0.99976 | 0.67 |
| Voltage | Transformer | 0.037 | 0.001 | 3.02578 | 2.98983 | 5095 |
| | Transformer$_b$ | 0.048 | 0.02 | 3.03208 | 2.98983 | 5080 |
| | RC | 0.004 | 1.00 | 2.98983 | 2.98983 | 1.4 |
| Ikeda map | Transformer | 0.219 | 0.00 | N/A | 1.0027 | 5100 |
| | Transformer$_b$ | 0.370 | 0.00 | N/A | 1.0027 | 5104 |
| | RC | 0.055 | 1.00 | 1.0073 | 1.0027 | 0.06 |

to an extreme value before stabilizing at a constant. For visualization clarity, this extreme value is omitted.

## 5.3 STATISTICS OF CRITICAL TRANSITION PREDICTION

The examples presented above are representative of the general behavior observed across our experiments, rather than isolated cases. To establish this conclusion more rigorously, we now present statistical evaluations. Across all trials, transformers never predicted a single collapse, regardless of configuration, even though their multi-step predictions in the training regime confirmed successful learning. Reservoir computing, in contrast, was capable of predicting collapse, though not deterministically: in few trials it produced oscillatory trajectories instead. Such variability is expected, as parameters only slightly above the critical threshold may not elicit sufficiently sensitive responses from every trained model.

A natural assumption is that a sufficiently expressive and well-trained model, when provided with a bifurcation parameter $p > p_c$, should eventually predict collapse if $p$ is gradually increased. To formalize this, we adopt the following criterion: starting from $p$ slightly above $p_c$, the parameter is increased step by step. For each value, the trained model generates a predicted trajectory. If the trajectory remains oscillatory below a certain $p$ but collapses beyond it, we define the corresponding parameter as the predicted critical point of that model. Based on this criterion, we conduct statistical experiments for both transformers and RC. Specifically, we independently train 50 models for each architecture. For each trained model, we generate 20 predictions with different warm-up segments, yielding $50 \times 20 = 1000$ simulations in total. We then record, within a predefined search range, the fraction of cases in which the model predicts collapse. We denote this probability as $P_c$(collapse). Notably, it is computed regardless of the predicted critical value itself, i.e., transformer may predict collapse only at a parameter much larger than the ground-truth $p_c$, but such cases are still counted as long as collapse occurs. In addition, it is worth noting that the search ranges differ between architectures: transformers, being less sensitive to parameter variations, are evaluated over a broader interval, whereas RC typically achieves near $100\%$ collapse prediction within a narrower range. Furthermore, we define collapse strictly as a dynamical drop of the observable variable below its oscillatory range. In cases with very large bifurcation parameters, the system may instead converge directly to a constant state without such a drop. These cases are not counted as successful collapse predictions.

The results are summarized in Tab. 1, which compares short-term forecasting accuracy in training and collapse prediction performance in testing across the three benchmark systems. RMSEs are reported as the mean error across the three training parameters, with multi-step forecasting horizons of 250, 250, and 10 steps for the food chain system, power system, and Ikeda map, respectively, corresponding to approximately $4 \sim 5$ Lyapunov times. Transformer$_b$ denotes the transformer variant that includes the same parameter-bias term used in RC (see Appendix B.2 for details). Although transformers require nearly two orders of magnitude more training data and more than 5000 seconds of training time, their RMSE values do not match the efficiency or accuracy of RC. Nevertheless, as illustrated in our examples, the multi-steps predictions by transformer are reasonable, as main-

taining accuracy for $4 \sim 5$ Lyapunov times is generally regarded as strong forecasting performance in nonlinear dynamics. In comparison, collapse prediction exposes a sharp divergence in performance. RC consistently achieves $100\%$ success and highly accurate estimates of the critical point, while transformers fail: the collapse prediction probabilities are only 0.38, 0.002, and 0.0 for the three systems. Moreover, even in the rare cases where transformers do predict collapse, the estimated critical points deviate substantially from the ground truth. This discrepancy is nontrivial. For example, as shown in Fig. 2(d), shifting the bifurcation parameter $K$ from 0.97 to 0.98 already traverses qualitatively different dynamics, so even seemingly small numerical errors correspond to severe dynamical mismatches. Further statistical analysis are provided in Appendix D.

## 6    CONCLUSION AND FUTURE DIRECTIONS

Transformers have achieved remarkable success in natural language processing, computer vision, and related domains, and have recently been proposed as foundation models for nonlinear dynamics. This work questions the effectiveness in this new setting. Rather than focusing on standard trajectory forecasting, we pose a challenge that is real-world important: critical transition and tipping prediction. The task requires models trained in regimes of periodic or chaotic oscillations, to predict under unseen parameters, that the system should collapse to extinction rather than maintain oscillations. To provide a baseline, we systematically compare transformers with RC across three benchmark systems: the food chain system, the power system, and the Ikeda map, and a spatiotemporal system: Kuramoto-Sivashisky equation.

The results are striking. While both transformers and RC achieve accurate multi-step forecasting in the safe parameter regime, their generalization diverges sharply under parameter drift. RC consistently anticipates collapse and locates the critical point with high precision. Whereas transformers, across all tested configurations, typically fail to predict collapse. In the rare cases where transformers succeed, the prediction arises only when the injected parameter value is excessively large. This raises a fundamental question: can transformers truly serve as faithful digital twins of dynamical systems? From a physics perspective, the performance gap stems from architectural principles. RC, being a dynamical system itself, naturally embeds target dynamics and generalizes across bifurcations. Transformers, relying on permutation-invariant self-attention, excel at sequence pattern extraction but lack inductive bias toward parameter-induced regime changes. It is worth emphasizing that our contribution does not lie in advocating RC as the ultimate solution for nonlinear dynamics. RC itself is limited by finite reservoir size and scalability constraints. Instead, our goal is to highlight an important and surprising comparison, raise the question of whether transformers capture dynamical principles, and suggest possible mechanisms behind their failure.

Beyond the results presented, prior studies show that RCs can reconstruct entire bifurcation diagrams, including transitions between periodicity and chaos, by training on a few parameter values (Kong et al., 2023). This explains strong extrapolation ability of RC in our experiments: it learns the underlying relationship between parameter and dynamics, rather than just trajectories. These observations also suggest examining whether transformers can similarly recover bifurcation structures within the safe regime, as a valuable direction. In addition, several related challenges remain open, such as building data efficient digital twins under limited observations Zhai et al. (2024a), and predicting not only the onset of collapse but also the post-collapse equilibrium. Incorporating physical constraints or structural priors may help address these gaps.

In language modeling, training defines a probability distribution over token sequences. In-distribution generalization represents generating new samples from this distribution, whereas out-of-distribution generalization corresponds to producing samples from a new distribution, given a few examples as context. In nonlinear dynamics, training on the pre-collapse attractor encourages a transformer to over-index on this specific distribution. Recent mechanistic studies show that transformers trained on a variety of synthetic sequence tasks often implement, or in-context learn, finite- or variable order Markov models (Ildiz et al., 2024; Edelman et al., 2024; Chen et al., 2024). This suggests that transformers may treat dynamical attractors as fixed sequence distributions, rather than reconstructing the underlying flow or its parameter dependence. A more detailed heuristic mechanistic discussion is provided in Appendix G.

Looking forward, several avenues merit exploration. One direction is the design of hybrid models, such as reservoir-attention architectures (Köster et al., 2024), that combine the dynamical embed-

ding ability of RC with the scalability of transformers to incorporate parameter-state relationships. Another is scaling: whether sufficiently large models, trained with diverse parameter regimes and domain-specific objectives, can eventually succeed remains an open question. Although foundation models do not perform well in system collapse prediction under zero-shot settings (see Appendix E), fine-tuning could possibly improve their performance. Exploring alternative parameter-conditioning mechanisms for transformers is also important; variants such as physics-informed (Zhao et al., 2023), continuous-time (Chen et al., 2023), or state-space (Gu & Dao, 2024) transformers may offer useful inductive biases. More broadly, physics-informed architectures and bifurcation-aware priors may be essential for robust digital twins of nonlinear systems. Reliable digital twins of nonlinear dynamics will play a significant role in forecasting critical events in ecology, climate, power grids, and beyond. We hope this work stimulates renewed attention to physics-guided architectures and inspires the development of next-generation machine learning frameworks that can faithfully capture not only the state evolution of a dynamical system but also how its behavior shifts, especially for critical transitions and tipping.

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

# A  ADDITIONAL BENCHMARK SYSTEMS

## A.1  POWER SYSTEM

We consider an electrical power systems including voltage collapse Dobson & Chiang (1989); Wang et al. (1994). The system exhibits transient chaos prior to collapse, making it an appropriate testbed for critical transition prediction. The model consists of four coupled differential equations, describing the rotor angle $\delta_m$, rotor speed $\omega$, load voltage phase $\delta$, and load voltage magnitude $V$:

$$
\begin{aligned}
\dot{\delta}_m &= \omega, \\
M\dot{\omega} &= -d_m\omega + P_m - E_m V Y_m \sin(\delta_m - \delta), \\
K_{qw}\dot{\delta} &= -K_{qv2}V^2 - K_{qv}V + Q(\delta_m, \delta, V) - Q_0 - Q_1, \\
TK_{qw}K_{pv}\dot{V} &= K_{pw}K_{qv2}V^2 + (K_{pw}K_{qv} - K_{qw}K_{pv})V \\
&\quad + K_{qw}[P(\delta_m, \delta, V) - P_0 - P_1] \\
&\quad - K_{pw}[Q(\delta_m, \delta, V) - Q_0 - Q_1],
\end{aligned}
\tag{4}
$$

where $V \angle \delta$ is the load voltage, $E_m \angle \delta_m$ is the generator terminal voltage, $E_0 \angle 0$ is the infinite bus voltage, and $\omega$ is the rotor angular speed. The load consists of a constant $PQ$ component in parallel with an induction motor. The real and reactive powers supplied to the load are

$$
P(\delta_m, \delta, V) = -E_0' V Y_0' \sin\delta + E_m V Y_m \sin(\delta_m - \delta), \tag{5}
$$

$$
Q(\delta_m, \delta, V) = E_0' V Y_0' \cos\delta - (Y_0' + Y_m)V^2 + E_m V Y_m \cos(\delta_m - \delta). \tag{6}
$$

Here, $Q_1$ represents the reactive power demand at the load bus and serves as the bifurcation parameter. As $Q_1$ increases, the system undergoes a boundary crisis bifurcation: for $Q_1 < Q_{1c}$, a periodic or chaotic attractor exists; for $Q_1 > Q_{1c}$, trajectories enter transient chaos and eventually collapse, i.e., the load voltage drops precipitously. The critical value is located at $Q_{1c} \approx 2.9898256$.

The constants are chosen following Kong et al. (2021a): $M = 0.01464$, $C = 3.5$, $E_m = 1.05$, $Y_0 = 3.33$, $\theta_0 = 0$, $\theta_m = 0$, $K_{pw} = 0.4$, $K_{pv} = 0.3$, $K_{qw} = -0.03$, $K_{qv} = -2.8$, $K_{qv2} = 2.1$, $T = 8.5$, $P_0 = 0.6$, $P_1 = 0.0$, $Q_0 = 1.3$, $E_0 = 1.0$, $Y_m = 5.0$, $P_m = 1.0$, $d_m = 0.05$. The adjusted Thévenin equivalents $(E'_0, Y'_0, \theta'_0)$ are defined in terms of $E_0$, $Y_0$, and $C$ as in (Kong et al., 2021a). The bifurcation diagram of the system is illustrated in Fig. 4(d). The narrow chaotic window makes the system especially sensitive to parameter drift and therefore an ideal benchmark for machine learning based digital twins.

## A.2 IKEDA MAP

We also consider the Ikeda map, which describes the dynamics of a laser pulse propagating through a nonlinear optical cavity (Ikeda et al., 1980; In et al., 1998). The map is defined on a complex variable $z \in \mathbb{C}$ and takes the form

$$z_{n+1} = \mu + \gamma z_n \exp\left(i\kappa - \frac{i\nu}{1 + |z_n|^2}\right), \tag{7}$$

where $\mu$ is the dimensionless laser input amplitude, $\gamma$ is the reflectivity coefficient of the cavity mirrors, $\kappa$ is the empty-cavity detuning, and $\nu$ characterizes detuning due to the nonlinear medium.

The Ikeda map has long served as a paradigmatic model for studying nonlinear optical dynamics, chaos, and crisis phenomena. Of particular interest is the occurrence of a boundary crisis bifurcation: for $\mu < \mu_c$, the system exhibits periodic or chaotic attractor, while for $\mu > \mu_c$, trajectories display transient chaos before escaping to a constant state, representing collapse. The critical point is located at $\mu_c \approx 1.0027$. In our setting, we follow the parameterization used in Kong et al. (2021b): $\gamma = 0.9$, $\kappa = 0.4$, and $\nu = 6.0$, while $\mu$ serves as the bifurcation parameter.

Beyond its relevance to nonlinear optics, the Ikeda map poses a stringent test for equation-discovery methods. Its update law involves a nested non-polynomial nonlinearity, $\exp(i\kappa - i\nu/(1 + |z|^2))$, for which a sparse representation in standard polynomial/Fourier libraries is generally unavailable when only observational time series are given. Consequently, sparse-regression approaches (e.g., SINDy-type methods) fail to recover a compact governing model, even though the dynamics are low-dimensional. This makes the Ikeda map an important case where black-box digital twins are necessary: reservoir computing succeeds in forecasting the state evolution and predicting the crisis, whereas equation discovery cannot.

## A.3 KURAMOTO-SIVASHINSKY SYSTEM

The Kuramoto-Sivashinsky (KS) system, as described by nonlinear partial differential equation (PDE), is a prototypical model for studying nonlinear spatiotemporal dynamics. Depending on the system parameters, its solutions range from regular traveling waves to sustained chaos and transient chaotic behavior, which makes the KS equation an informative benchmark for evaluating whether a machine learning model can capture qualitative changes in dynamics when a bifurcation parameter is varied. In this work, we consider the one-dimensional KS equation on a periodic domain $x \in [0, \pi]$:

$$\frac{\partial u}{\partial t} + \nu \frac{\partial^4 u}{\partial x^4} + \phi \left(\frac{\partial^2 u}{\partial x^2} + u \frac{\partial u}{\partial x}\right) = 0, \tag{8}$$

where $u(x, t)$ is a scalar field and $\nu$, $\phi$ are system parameters. We fix $\nu = 4$ following standard practice for generating transient chaos and treat $\phi$ as the bifurcation parameter (Kong et al., 2021a).

For $\phi$ below a critical value $\phi_c \approx 200.04$, the system exhibits sustained spatiotemporal chaos characterized by irregular fluctuations in both space and time. When $\phi$ is increased past $\phi_c$, the chaotic attractor is destroyed through a crisis-like transition: trajectories enter a regime of transient chaos and eventually collapse onto a stable traveling-wave solution. This collapse state is never seen during training, since all training trajectories are generated at parameter values strictly below $\phi_c$.

Table 2: Transformer hyperparameters

| Hyperparameter | Value |
| --- | --- |
| Embedding dimension $N$ | 128 |
| Hidden size (FNN) | 256 |
| Number of layers $N_b$ | 4 |
| Number of attention heads | 4 |
| Dropout rate | 0.2 |
| Maximum sequence length $L_{\max}$ | 512 |
| Noise level (training) | 0.05 |
| Total parameters | $\sim 5 \times 10^6$ |

## B  MACHINE LEARNING METHODS

### B.1  TRANSFORMER

We adopt a decoder-only transformer architecture with a causal attention mask, so that each state depends only on its past. The input multivariate time series $X \in \mathbb{R}^{L \times d}$, where $L$ is the sequence length and $d$ the variable dimension, is first projected through a linear layer to an embedding dimension $N$. To preserve temporal ordering, sinusoidal positional encodings are added to the embeddings before entering the transformer blocks. Each transformer block processes a hidden representation $H^{(\ell)} \in \mathbb{R}^{L \times N}$, where $\ell$ denotes the layer index. Within each block, queries, keys, and values are computed as

$$Q^{(\ell)} = H^{(\ell)} W_Q^{(\ell)}, \quad K^{(\ell)} = H^{(\ell)} W_K^{(\ell)}, \quad V^{(\ell)} = H^{(\ell)} W_V^{(\ell)}, \tag{9}$$

and the scaled dot-product attention is

$$\text{Attention}(Q^{(\ell)}, K^{(\ell)}, V^{(\ell)}) = \text{softmax}\left(\frac{Q^{(\ell)} K^{(\ell)\top}}{\sqrt{d_k}}\right) V^{(\ell)}. \tag{10}$$

Multiple heads are computed in parallel, concatenated, and linearly projected, after which residual connections, layer normalization, and a position-wise feed-forward network with ReLU activation are applied. The resulting $H^{(\ell+1)}$ serves as the input to the next block. Stacking $N_b$ such blocks enables the model to capture long-range temporal dependencies.

At the output layer, a linear projection maps the final hidden representation back to the original dimension $d$, yielding the next-step prediction $\hat{X}$. Multi-step forecasts are obtained in closed loop by feeding predictions back into the model. To account for parameter-induced behavior shifts, we extend the input with an additional channel containing the bifurcation parameter. The hyperparameters are identical for the three benchmark systems and are listed in Tab. 2.

### B.2  RESERVOIR COMPUTING

RC is a recurrent neural network framework particularly well suited for modeling dynamical systems. Its main idea is to embed the input time series into a high-dimensional dynamical network with fixed random connections, while only training a linear readout layer. To capture parameter-induced regime changes, we employ a parameter-aware reservoir with an additional input channel for the bifurcation parameter, following the general framework of Kong et al. (2021a;b). Given an input state $X(t) \in \mathbb{R}^d$ from the target system and a bifurcation parameter $p$, the reservoir state $r(t) \in \mathbb{R}^n$ evolves according to

$$r(t + \Delta t) = (1 - \alpha)r(t) + \alpha \tanh(W_r r(t) + W_{\text{in}} X(t) + W_p(p + p_0)), \tag{11}$$

where $W_r \in \mathbb{R}^{n \times n}$ is the fixed recurrent weight matrix, $W_{\text{in}} \in \mathbb{R}^{n \times d}$ projects the system input into the reservoir, and $W_p \in \mathbb{R}^{n \times 1}$ injects the bifurcation parameter $p$ with parameter bias $p_0$. The leakage rate $\alpha \in (0, 1]$ controls the update rate. The nonlinear activation is the hyperbolic tangent.

Table 3: Reservoir computing hyperparameters

| Hyperparameter | Food chain | Power system | Ikeda map | Kuramoto-Sivashinsky |
|---|---|---|---|---|
| Reservoir size $n$ | 900 | 800 | 400 | 1000 |
| Link probability $P_L$ | 0.004 | 0.313 | 0.708 | 0.450 |
| Spectral radius $\rho$ | 2.3 | 1.6 | 0.17 | 0.89 |
| Input scaling $k_{\text{in}}$ | 3.6 | 2.1 | 2.6 | 0.057 |
| Parameter scaling $k_p$ | 0.5 | 1.6 | 0.35 | -0.052 |
| Parameter bias $p_0$ | $-2.2$ | $-3.1$ | 0.47 | -185 |
| Leakage rate $\alpha$ | 0.30 | 1.0 | 1.0 | 1.0 |
| Regularization $\beta$ | $3 \times 10^{-5}$ | $1 \times 10^{-4}$ | $1 \times 10^{-6}$ | $8 \times 10^{-5}$ |

The reservoir output is a linear readout of the hidden state,

$$\hat{X}(t) = W_{\text{out}} r(t), \tag{12}$$

where $W_{\text{out}} \in \mathbb{R}^{d \times n}$ is trained by ridge regression to minimize

$$\mathcal{L} = \sum_t \|X(t) - W_{\text{out}} r(t)\|^2 + \beta \|W_{\text{out}}\|^2, \tag{13}$$

with $\beta > 0$ a regularization parameter. During training, ground-truth trajectories drive the reservoir; during prediction, the output is fed back, i.e., $X(t) \leftarrow \hat{X}(t)$. This closed-loop operation turns the trained RC into a self-evolving dynamical system that, for a given parameter $p$, generates trajectories in the corresponding regime. As RC performance is sensitive to hyperparameters, we use Bayesian optimization during validation. The optimized hyperparameters are listed in Tab. 3.

It is worth noting that, for both RC and transformers, we use the same $(d + 1)$-dimensional input: the $d$-dimensional state concatenated with a parameter channel. In RC, the input terms in Eq. (11) can be written as

$$W_{\text{in}} X(t) + W_p p = [W_{\text{in}}, W_p] [X(t); p], \tag{14}$$

if we remove parameter bias $p_0$, which shows that the parameter is simply appended as the last channel. Here $W_{\text{in}}$ and $W_p$ are written separately because they are drawn from different ranges to match the scaling of states and parameter. In the transformer, the parameter channel is appended in exactly the same way, and its input weights are adjusted during training. In addition, to make the comparison fair, we also include a parameter bias for the transformer, as $p_0$ in RC. The results are listed as transformer$_b$ in Table 1.

## C  FURTHER DEMONSTRATION OF CRITICAL TRANSITION PREDICTION

### C.1  POWER SYSTEM

In this section we demonstrate critical transition prediction of the voltage collapse in power systems. As introduced in Appendix A.1, this system is four-dimensional and exhibits transient chaos preceding collapse as the load parameter $Q_1$ increases. The critical value is located near $Q_{1c} \approx 2.9898256$. During training, trajectories are generated at $Q_1 = [2.98968, 2.98973, 2.98978]$, all within the safe chaotic regime. Testing is performed at $Q_1 = 2.989830 > Q_{1c}$, where the ground truth shows a short chaotic transient followed by a sudden voltage drop.

Figure 4 compares the transformer and RC predictions. Consistent with the main text, both models perform reliably on the training parameters. Afterward, they are evaluated under a parameter shift beyond the critical point. The transformer, while accurate on training trajectories, do not exhibit sensitivity to parameter shifts to produce collapse behavior. By contrast, RC correctly follows the transient oscillations and then reproduces the collapse, in agreement with the ground truth dynamics.

### C.2  IKEDA MAP

We also consider the Ikeda map, a paradigmatic nonlinear optical system describing the evolution of a laser pulse in a nonlinear cavity, as detailed in Appendix A.2. It is a discrete-time, one-dimensional

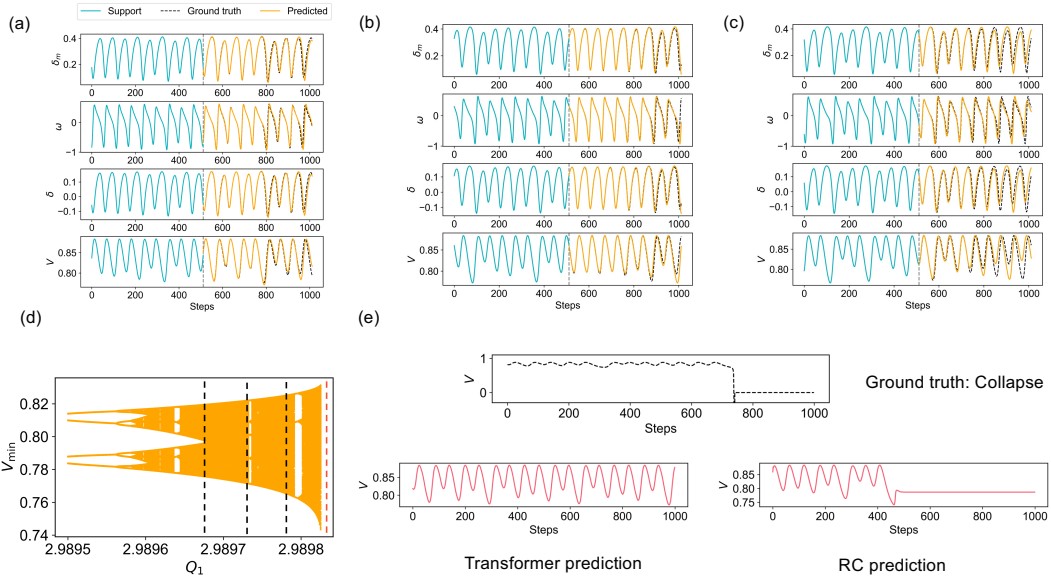

Figure 4: Critical transition prediction in the power system. (a–c) Blue curves indicate warm-up input, and orange curves denote transformer predictions at the three training parameters $Q_1$. (d) Bifurcation diagram with training parameters (black dashed) and the testing parameter beyond the critical point (red dashed). (e) Comparison of transformer and RC predictions at $Q_1 = 2.989830 > Q_{1c}$. The ground truth trajectory collapses after a transient; RC reproduces this collapse, whereas the transformer continues oscillating.

complex map that undergoes a boundary crisis at $\mu_c \approx 1.0027$. Training is carried out in the safe regime at $\mu = [0.91, 0.94, 0.97]$, while testing is performed at $\mu = 1.01 > \mu_c$, where the ground truth dynamics collapse.

Figure 5 presents the results. As in previous systems, the transformer fails to recognize the collapse and instead generates oscillatory trajectories similar to those seen during training. RC, in contrast, successfully captures the transient behavior and predicts the eventual collapse. The Ikeda case is particularly noteworthy: because its nonlinear update rule cannot be represented in a sparse polynomial or Fourier library, sparse-regression approaches fail. As a result, black-box digital twins are essential in this setting. RC proves effective, while transformers again fall short.

## C.3 KURAMOTO-SIVASHINSKY SYSTEM

We consider KS equation, a high-dimensional nonlinear spatiotemporal system that was originally derived to model instabilities in laminar flame fronts, and later applied to other physical systems, such as trapped-ion instability in plasmas (LaQuey et al., 1975).A boundary crisis occurs at $\phi_c \approx 200.04$. Training is performed in the safe regime at three parameter values $\phi \in \{196, 197, 198\}$, where all trajectories remain chaotic. Testing is conducted at a supercritical parameter $\phi = 200.14 > \phi_c$, at which the ground truth collapses after a transient.

Figure 6 demonstrates representative results. Panels (a-c) show representative results, where both the transformer and RC can accurately reproduce the KS chaotic dynamics in the training regime. When evaluated at $\phi = 200.14$, however, the two models behave differently: as shown in Fig. 6(e), RC successfully captures the transient chaotic phase and the subsequent collapse into a traveling-wave state, whereas the transformer fails and continues to generate chaotic predictions indefinitely.

It is important to note that the collapse behavior in the KS system differs fundamentally from the other benchmark systems. In those ODE examples, collapse corresponds to a state variable abruptly falling to a fixed point far away from the original attractor. In contrast, the KS system transitions from broadband spatiotemporal chaos to a nearly monochromatic traveling-wave solution.

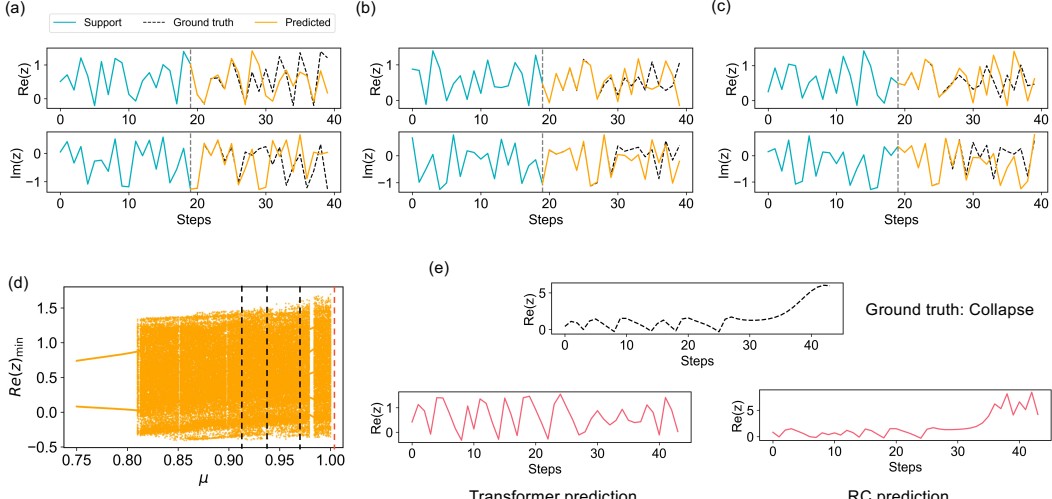

Figure 5: Critical transition prediction in the Ikeda map. (a–c) Blue curves indicate warm-up data, and orange curves denote transformer predictions at the three training parameters $\mu$. (d) Bifurcation diagram with training parameters (black dashed) and the testing parameter beyond the critical point (red dashed). (e) Comparison of transformer and RC predictions at $\mu = 1.01 > \mu_c$. RC reproduces the collapse after a transient, whereas the transformer fails and remains oscillatory.

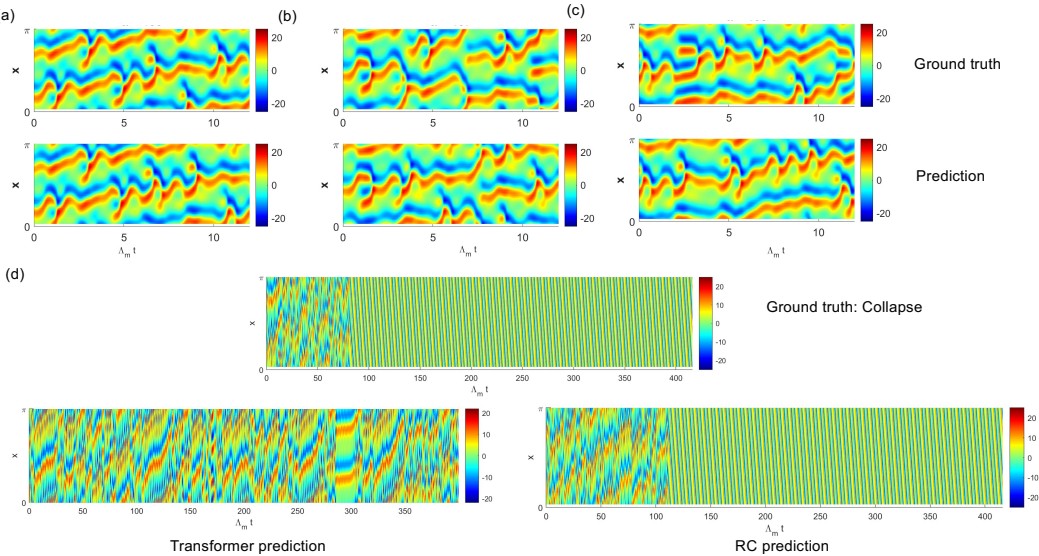

Figure 6: Critical transition prediction in the Kuramoto-Sivashinsky system. (a–c) Upper panels indicate ground truth, and lower denote representative predictions at the three training parameters $\phi$. (e) Comparison of transformer and RC predictions at $\phi = 200.14 > \phi_c$. RC reproduces transition from chaos to a traveling wave, whereas the transformer remains in a chaotic state.

To quantify this transition, we analyze the temporal signal extracted at a representative spatial location (e.g., the midpoint of the domain). For each sliding time window of length $L_w$, we denote the windowed segment by $u(t)$ and compute its discrete Fourier transform, $U_k = \text{FFT}(u(t))$. The corresponding power spectrum is $P_k = |U_k|^2$. We then define the spectral concentration ratio

$$R = \frac{\max_{k>0} P_k}{\sum_{k>0} P_k}, \tag{15}$$

Table 4: Performance on critical transition prediction on Kuramoto-Sivashinsky system

| system | model | $P_c$(collapse) $\uparrow$ | pred. crit. | gt crit. | $T_{\mathbf{train}}[\mathbf{s}]\downarrow$ |
|--------|-------|---------------------|-------------|----------|------------------|
| KS | Transformer | 0.30 | 200.332 | 200.04 | 8500 |
|    | RC | 0.94 | 200.163 | 200.04 | 2.29 |

where the sum and maximum are taken over all nonzero positive frequencies. In the chaotic regime, the spectrum is broadband and $R$ is relatively small. After collapse, the traveling-wave solution exhibits a more concentrated spectrum, and $R$ becomes significantly larger.

Let $R_i$ denote the ratio computed from the $i$-th sliding window, and let $\widetilde{R}_i$ be a short moving-average smoothing of $\{R_i\}$. A collapse is declared if

$$\widetilde{R}_i > \theta_R \quad \text{for at least } m \text{ consecutive windows,} \tag{16}$$

where we set the threshold to $\theta_R = 0.5$. To avoid false positives, we additionally require that the majority of windows prior to the transition satisfy $\widetilde{R}_j < \theta_R$, ensuring that the trajectory was chaotic before becoming spectrally concentrated. The collapse time is then defined as the center time of the first window fulfilling these conditions.

Table 4 summarizes the statistical results. Both RC and transformer require longer training time on this high-dimensional PDE system. The transformer performs poorly even after extending the sequence length to $L_{\max} = 1024$ and downsampling the data (every two points) to fit memory constraints. RC achieves a collapse prediction rate of $94\%$, which is lower than for the ODE benchmarks but remains substantially higher than the transformer.

## D ADDITIONAL STATISTICS OF CRITICAL TRANSITION PREDICTION

In the main text, we summarized the statistical performance of transformer and RC in critical transition prediction in Table 1. To provide a more detailed view, we depict histograms of the predicted critical points across all trials in Fig. 7. For each system and each model, we conducted 50 independent trainings, with 20 predictions per training (different warm-up segments), yielding 1000 realizations in total. Specifically, the histograms in Fig. 7 show the distribution of predicted critical values $\hat{p}_c$ compared with the ground-truth $p_c$. The first row corresponds to transformers and the second to RC. The three columns represent the food chain system, power system, and Ikeda map, respectively. In each panel, the red dashed line indicates the ground-truth $p_c$, while the black dashed line marks the mean of the predicted values across all realizations. For the transformer, in the power system only a single collapse prediction was observed among 1000 runs, we take that as the mean and depict as dashed black line in panel (b). For the Ikeda map, no collapse was predicted in any realization, and therefore only the ground-truth line is shown in panel (c). The horizontal axis in each histogram matches the parameter search range used in the experiments; values outside this range are omitted as they lie far beyond the true critical point.

These results further highlight the contrast between the two models. For transformers, collapse predictions are nearly absent in the power system and Ikeda map, and even in the food chain system, where collapse was predicted in $38\%$ of realizations, the averaged $\hat{p}_c$ values were significantly biased upward relative to $p_c$. In contrast, RC achieves robust and reliable results: not only did it consistently predict collapse in all realizations, but the predicted critical points were tightly concentrated around the ground truth.

## E FOUNDATIONS MODELS ON SYSTEM COLLAPSE PREDICTION

Recent advances in time series modeling have produced pretrained foundation models that demonstrate strong performance across forecasting, anomaly detection, and imputation tasks (Lai et al., 2025; Ansari et al., 2024; Das et al., 2024; Liang et al., 2024). Of particular relevance to nonlinear dynamics are models trained directly on large collections of chaotic systems, which aim to

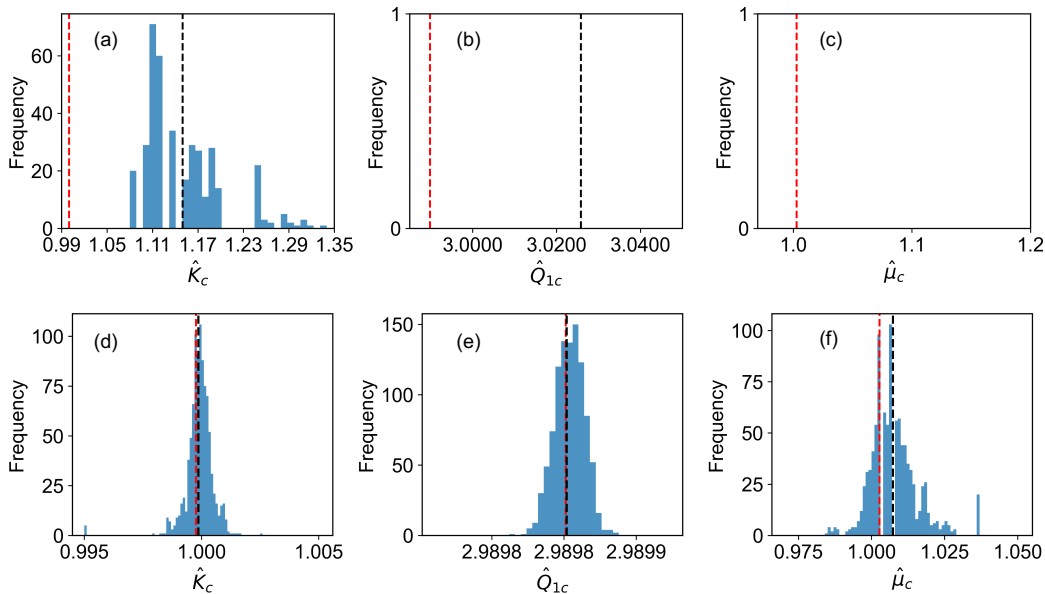

Figure 7: Histograms of predicted critical points $\hat{p}_c$. Rows correspond to models (top: transformer, bottom: RC). Columns correspond to systems: (a,d) Food chain system. (b,e) Power system, and (c,f) Ikeda map. Red dashed line denote the ground-truth critical points, where $K_c = 0.99976, Q_{1c} = 2.9898256$, and $\mu_c = 1.0027$. Black dashed line indicate the mean predicted critical points. $\langle \hat{K}_c \rangle = 1.14958$ and $\langle \hat{Q}_{1c} \rangle = 3.02578$ for the transformer, and $\langle \hat{K}_c \rangle = 0.99986$, $\langle \hat{Q}_{1c} \rangle = 2.98983$ and $\langle \hat{\mu}_c \rangle = 1.0073$ for RC.

generalize across dynamical regimes in a zero-shot manner (Lai et al., 2025). In this section, we evaluate two representative models on the problem of system collapse prediction: PANDA, a pretrained model for chaotic dynamics, and Chronos-2, a multivariate foundation model for time-series forecasting.

### E.1 PANDA

PANDA is a pretrained forecast model specifically designed for nonlinear and chaotic dynamics (Lai et al., 2025). It is trained on an extensive synthetic dataset of more than $2 \times 10^4$ chaotic ODE systems, generated through evolutionary recombination of known dynamical systems. PANDA utilizes a patched attention architecture, enriched with dynamics-motivated embeddings, such as polynomial and Fourier features. PANDA demonstrates impressive zero-shot forecasting capabilities on unseen chaotic ODEs and even PDEs.

However, its predictions can deteriorate in long-term chaotic attractor reconstructions, where it could drift toward fixed points. Because system collapse prediction requires accurate long-horizon forecasting within the safe regime as a prerequisite, such drift limits its suitability for our task. Figure 8 illustrates representative examples of PANDA zero-shot predictions for the food chain system. PANDA predicts several cycles with high accuracy in both safe and collapse regimes, but cannot reconstruct attractors or detect collapse reliably in the long term.

### E.2 CHRONOS-2

Chronos-2 Fatir Ansari et al. (2025) is the recently released multivariate extension of the Chronos family, designed as a general foundation model for time series forecasting. It incorporates cross-channel attention and can process multivariate trajectories, making it applicable to our tasks. The model is trained using large-scale probabilistic objectives on diverse real-world datasets, and it achieves strong short-term predictive performance in many conventional forecasting tasks.

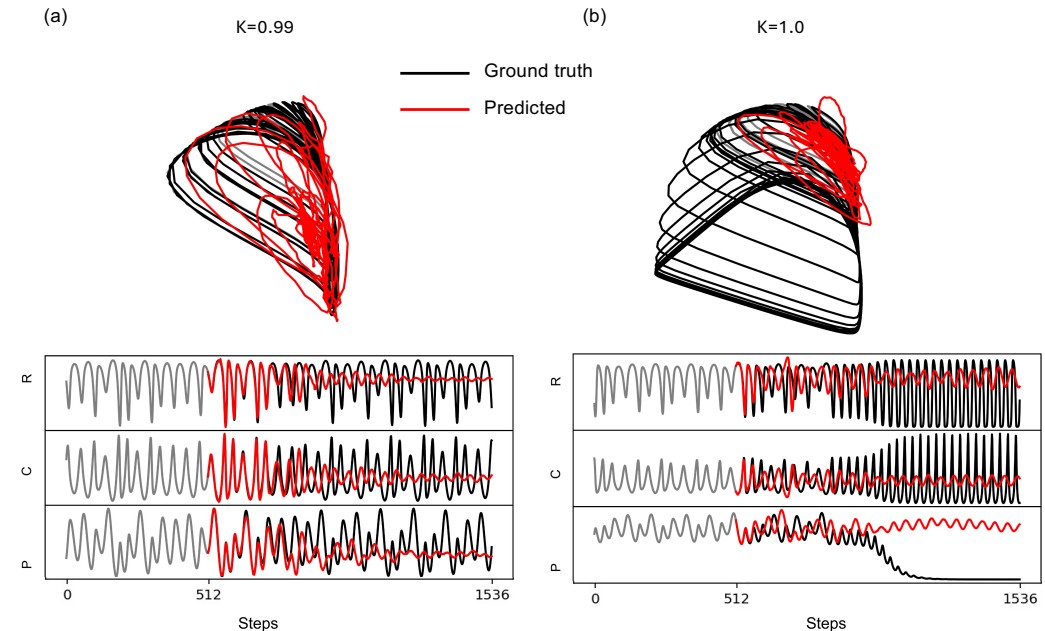

Figure 8: Food chain system predictions by PANDA. (a) Long- (upper) and short-term (lower) predictions under safe regime. (b) Long- (upper) and short-term (lower) predictions under collapse. PANDA provides accurate zero-shot predictions for several cycles but fails to maintain long-term accuracy.

For chaotic dynamics, Chronos-2 shows reasonable performance, but worse than PANDA in short-term prediction, which is expected given that PANDA is specifically trained on chaotic systems. Furthermore, like other time series foundation models, it fails to preserve attractor geometry during long predictions. As shown in Fig. 9, Chronos-2 cannot reconstruct chaotic dynamics in the safe regime and does not anticipate parameter-induced collapse.

## F    FURTHER EXPERIMENTS ON TRANSFORMERS

Transformers are well known to be data-hungry and prone to overfitting. In our main experiments, we demonstrated that for RC, training on trajectories from only three bifurcation parameters was already sufficient to infer unseen regimes with collapse. For fairness, the same training setting was adopted for transformers. However, with such limited data, transformers, owing to their quite large number of trainable parameters, may simply memorize the trajectories associated with each training parameter, rather than learning the underlying dynamical rules. This raises the concern that their apparent accuracy in the training regime may reflect overfitting rather than genuine generalization.

To study this, we perform in this section, additional experiments with denser bifurcation parameter sampling. For each benchmark system, instead of using only three bifurcation parameters, we generated much larger training sets: 27, 21, and 31 parameter values for the food chain system, power system, and Ikeda map, respectively. These cover the similar parameter ranges as in the main text. The shaded blue intervals in Fig. 10(a–c) denote the expanded training ranges, with black dashed lines indicating the parameters used previously and the red dashed line marking the test parameter beyond the critical point.

Figure 10(d) shows representative transformer results in the food chain system. Across a wide variety of training parameters, the model achieves accurate multi-step forecasts, confirming that the network is well trained. Panels (e–g) further illustrate transformer predictions at supercritical parameters for the food chain system, power system, and Ikeda map, respectively. In all cases, although the models are well trained in the safe regimes, they continue to produce oscillatory trajectories instead of reproducing collapse. In other words, even with substantially more training data, transformer

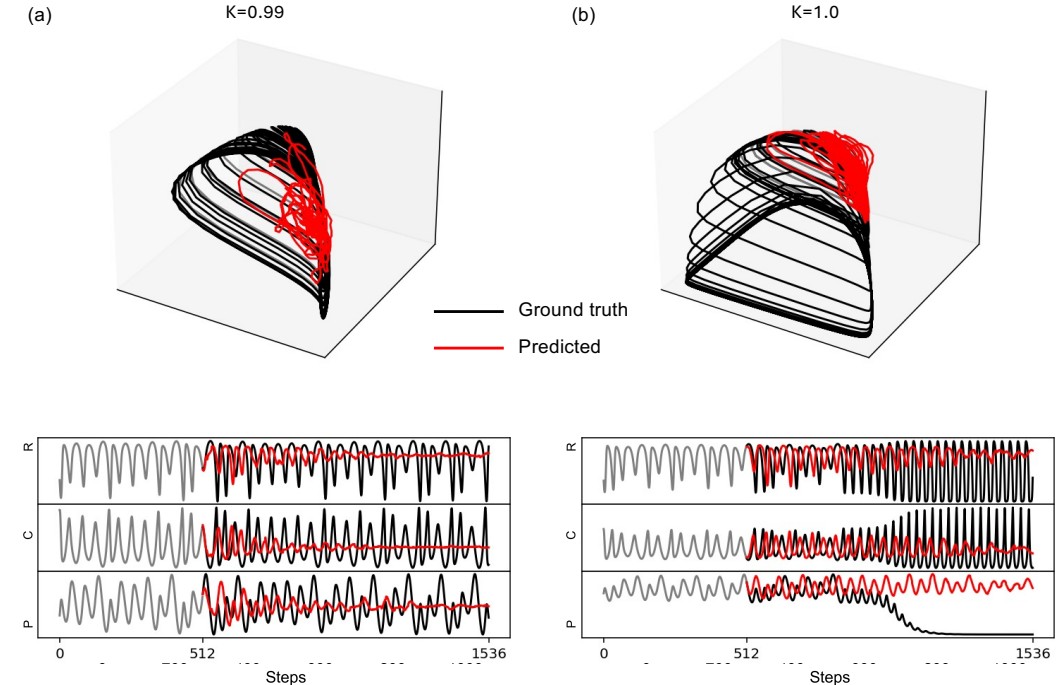

Figure 9: Food chain system predictions by Chronos-2. (a) Long- (upper) and short-term (lower) predictions under safe regime. (b) Long- (upper) and short-term (lower) predictions under collapse. Chronos-2 shows limited short-term accuracy and fails in long-term forecasting.

performance on critical transition prediction does not improve in any significant way. By contrast, RC achieves high success rates in collapse prediction using only three training parameters, without requiring such extensive data.

In addition, to further study transformer performance, we conduct experiments under multiple architectural configurations. Specifically, we vary the embedding dimension, hidden size, number of layers, and number of attention heads. Three representative configurations were selected as [128, 64, 2, 2], [256, 256, 4, 4] and [512, 256, 8, 8], corresponding to models with approximately $6.7 \times 10^5$, $2.6 \times 10^6$, and $1.5 \times 10^7$ trainable parameters, respectively. For simplicity, we denote these as Models #1, #2, and #4. Model #3 refers to the architecture used in the main text (see Tab. 2). Figure 11 demonstrates multi-step forecasting results averaged over 10 independent realizations. Standard deviations are relatively large, reflecting both the variability across independently trained models and differences across training bifurcation parameters within the same model. Among the four configurations, Model #3 yields the best multi-step forecasting performance and is therefore adopted as our main transformer baseline. Moreover, additional tests are conducted, which confirm that none of the alternative configurations improved performance on critical transition prediction.

## G HEURISTIC MECHANISTIC ANALYSIS

We provide a heuristic comparison between parameter-aware RC and parameter-aware transformers in the context of constructing a digital twin for nonlinear dynamical systems.

RC relies on its hidden reservoir state $r(t)$, which evolves according to Eq. (11), based on both its previous state and the current input. This makes RC itself a driven dynamical system, excited by the target system state and parameter $(x(t), p(t))$. Under the echo-state property (ESP), the driven reservoir converges to a unique trajectory determined by the input history Jaeger et al. (2007). In this regime, the reservoir establishes a form of generalized synchronization: $r(t)$ becomes an approximate high-dimensional embedding of the underlying attractor for each parameter value.

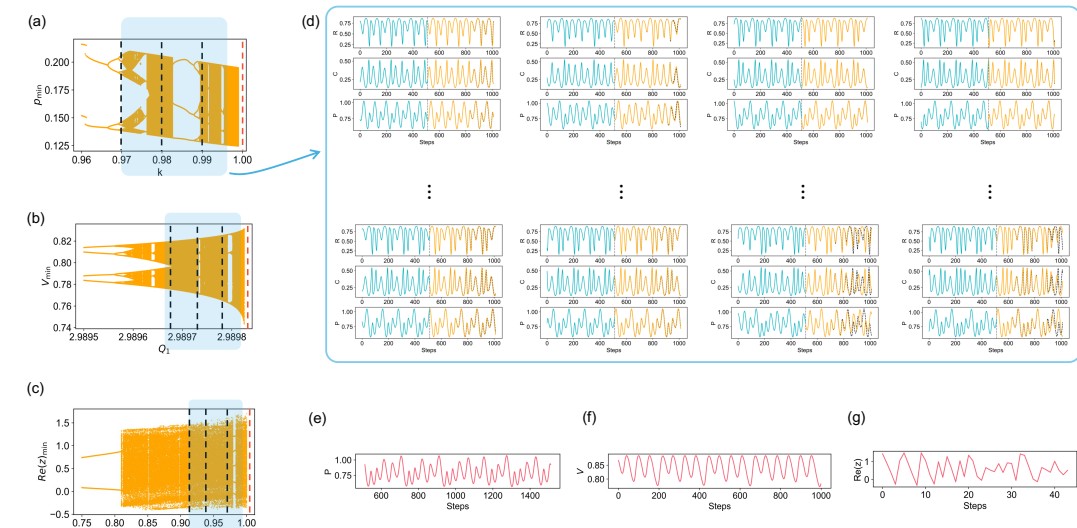

Figure 10: Effect of increased training data on transformer. (a-c) Expanded training parameter ranges (blue shaded), with parameters used in the main text (black dashed) and the supercritical test parameter (red dashed). (d) Example transformer predictions for the food chain system, trained with 27 parameter values. (e-g) Transformer critical transition predictions for the food chain system, power system, and Ikeda map, respectively. Although the model performs well across many safe parameters, it continues to generate oscillations rather than collapse beyond the critical point.

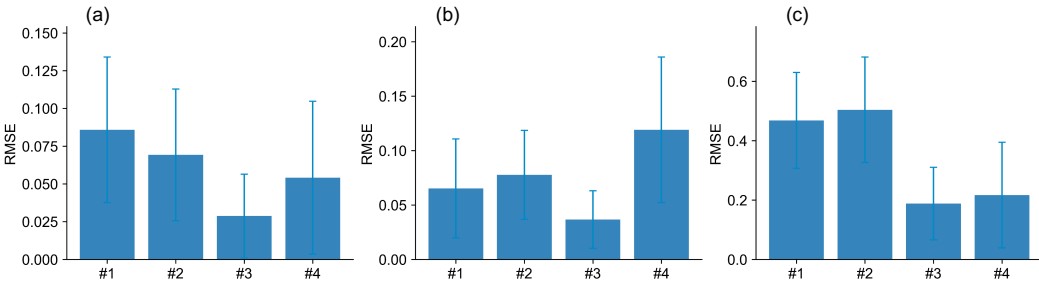

Figure 11: Multi-step forecasting accuracy of different transformer configurations. Error bars denote standard deviations across 10 independent realizations. Model #3 corresponds to the baseline used in the main text.

Importantly, after training, RC does not simply integrate information within a finite history window. Instead, it evolves autonomously, on a learned manifold with $p$ acting as a continuous control input under the parameter-aware design. Small variations in $p$ therefore move the driven reservoir state smoothly along a high-dimensional manifold that mirrors how the physical attractor deforms with parameter. This induces a strong inductive bias: when a boundary crisis occurs and the physical attractor disappears, the reservoir asymptotic dynamics naturally change as well. Thus, even without being trained directly "on the bifurcation," RC inherits sensitivity to global stability structure and can fall off the embedded chaotic manifold into the collapsed state for the corresponding parameter.

In contrast, a vanilla transformer with causal attention implements a static mapping $(x, p)_{t-L+1:t} \mapsto \hat{x}_{t+1}$, where history enters only through self-attention over a finite window of length $L_{\max}$. Self-attention is fundamentally permutation-invariant. Although positional encodings reintroduce temporal order, the architecture remains a feed-forward function from a finite sequence to the next step; it lacks an internal state governed by a dynamical law, and thus do not change the fact that self-attention lacks built-in notions of stability, phase space, or bifurcations. As a result, transformers primarily learn sophisticated autoregressive rules: mappings from recent oscillatory patterns to the

next value. This is sufficient to reproduce accurate short-term forecasts, and even approximate the invariant measure of the attractor within the training (safe) regime, but it does not require reconstruction of a global state space or vector field. Consequently, the parameter channel $p$ could be often learned only as a mild modulation of amplitude or frequency, rather than as a control capable of shifting or destroying attractors.

Critical transitions, however, are governed by global bifurcations such as boundary crises, which depend sensitively on the geometry of stable and unstable manifolds. Capturing them requires, at least approximately, embedding the basin boundary and tracking how it moves with $p$. RC naturally acquires this through generalized synchronization and the ESP. Transformers, despite powerful pattern-extraction ability, lack mechanisms to encode unstable manifolds or Lyapunov structure. They fit local temporal correlations on the attractor but fails in learning how global stability depends on $p$. As a result, for $p > p_c$, the transformer closed-loop dynamics tend to remain locked onto an "effective" attractor inherited from the safe regime, even though the true attractor has already disappeared.

