# OpenReview forum: "Can transformers truly understand dynamical systems?"
_ICLR.cc/2026/Conference — Submitted to ICLR 2026_

### Official Review · Reviewer_aMYQ · 2025-10-17

**Soundness:** 2
**Presentation:** 2
**Contribution:** 2
**Rating:** 4
**Confidence:** 5

**Summary:**

The authors show that reservoir computing can anticipate critical transitions and tipping points while transformers cannot, casting doubt on transformers' generalization ability in learning dynamical systems.

**Strengths:**

Understanding the strengths and weaknesses of the transformer architecture in learning dynamical systems is an important problem. Anticipating tipping points is also a challenging task that requires the ML model to go beyond its training data, with many potential practical applications.

**Weaknesses:**

* The authors only study one specific task: anticipating critical transitions. The title "Can transformers truly understand dynamical systems?" is too grand for the actual content of the paper and might mislead the readers.
* All experiments only included three toy systems, which feels a bit thin to support the strong conclusions.
* There is no theoretical insights into why transformers fundamentally cannot capture critical transitions. Without such insights, it is unclear whether transformers failed because it is not a suitable architecture, or because the authors' implementation is suboptimal.

**Questions:**

* In parameter-aware RC, the bifurcation parameters are used to drive the reservoir state. However, there is no such internal dynamics in transformers. How did the authors use the bifurcation parameters to "drive" the transformers? It is possible that the transformers failed simply because the bifurcation parameters weren't utilized effectively in this particular setup.
* Anticipating tipping points is a difficult task that requires out-of-distribution generalization. Can you provide a theory to explain why RC can successfully extrapolate and why transformers cannot?
* I got the impression that RC also cannot reliably predict the collapse time and collapsed states of a system past the critical transition point, so in this sense RC also does not truly "understand" dynamical systems, right?
* It was mentioned that "One direction is the design of hybrid models, such as reservoir-attention architectures, that combine the dynamical embedding ability of RC with the scalability of Transformers." What do authors mean exactly by the reservoir-attention architecture?
* The authors keep saying that transformers may not be suitable for dynamical systems because the attention mechanism is permutation invariant. But didn't positional encoding address this problem?

---

> ### Author Response · Authors · 2025-11-21
>
> We thank the referee for the constructive and insightful feedback, and for recognizing the importance and potential impact of this line of research. Our responses and corresponding revisions are as follows.
>
> 1. We agree with the referee that the title should be more specific, we have changed it into ``Can transformers predict system collapse in dynamical systems?''
>
> 2. Although all dynamical systems are abstraction, we intentionally selected benchmarks that span different scientific domains and modeling characteristics. The Ikeda map is a paradigmatic nonlinear optical model with a non-polynomial structure; the food chain system is a three-dimensional ecological model with a bioenergetic interpretation; and the voltage collapse model is four-dimensional and directly linked to real power grid in stability analysis. In the revised paper, we further include a spatiotemporal PDE system (Kuramoto-Sivashinsky) to enrich our experiments (Appendix A.3 and Appendix C.3).
>
> 3. We added Appendix G to provide a heuristic analysis explaining why RC can extrapolate across bifurcations while transformers fail. We also expanded experiments to include additional transformer configurations and pretrained foundation models (Table 1 and Appendix E).
>
> Q1. We appreciate the referee for the helpful insight. Both RC and transformer models receive $(d+1)$-dimensional input: the $d$-dimensional state together with one parameter channel. RC uses this parameter to drive its internal recurrent dynamics, while transformers have no inherent dynamical state that evolves with the parameter. This architectural mismatch is exactly why the transformer fails to use bifurcation parameters effectively. Exploring alternative parameter conditioning mechanisms for transformers is a promising future direction. We have included this discussion in Conclusion and Future Directions, in line 490-493.
>
> Q2. A detailed heuristic explanation is provided in Appendix G.
>
> Q3. This is a great question. In nonlinear dynamical systems, even when the governing equations and parameters are fixed, small variations in initial conditions generally lead to different collapse times and different post-collapse equilibria, due to the underlying basin structure. Despite this intrinsic variability, RC consistently predicts whether collapse will occur and accurately identifies the collapse parameter, whereas transformers often predict that collapse will not occur at all. This qualitative difference is practically important.
>
> Q4. By ``reservoir-attention architectures,'' we refer to recent approaches that integrate recurrent reservoir dynamics with attention blocks, combining the expressive power of RC with the scalability of transformers. Such hybrid models may help transformers to incorporate parameter-state relationships that plain attention cannot capture. We have clarified the description in line 485-487.
>
> Q5. Positional encodings (PE) reintroduce order information but do not change the fact that self-attention is fundamentally permutation-invariant and lacks built-in notions of stability, phase space, and bifurcations. PE alleviates but does not eliminate this structural mismatch, which is critical in regime shift prediction. We added this explanation in Appendix G.

---

### Official Review · Reviewer_L6LU · 2025-10-27

**Soundness:** 2
**Presentation:** 3
**Contribution:** 2
**Rating:** 4
**Confidence:** 4

**Summary:**

This study evaluates whether Transformer architectures can function as faithful digital twins of nonlinear dynamical systems. Using a benchmark task of predicting catastrophic collapse when bifurcation parameters cross critical thresholds, the authors train Transformers and reservoir computing (RC) models on time series generated from safe parameter regimes, and test them on unseen regimes past the point at which the dynamics collapse. Across three nonlinear systems (food chain, power grid, and Ikeda map), Transformers achieve strong short-term forecasts but consistently fail to predict state collapse, instead producing sustained oscillations. In contrast, RC models reliably anticipate critical transitions with high accuracy and minimal data. The findings suggest that permutation-invariant self-attention lacks sensitivity to parameter-induced regime shifts, in contrast to the intrinsic dynamical embedding of RC. This work aims to challenge the claimed generalization capabilities of Transformers for dynamical systems, and proposes critical-transition prediction as a benchmark for assessing digital twin fidelity

**Strengths:**

**Motivation.** This is a great idea for a paper. Isolating generalization in dynamical systems by studying cessation of chaotic dynamics makes sense. In language modeling, we often see the training data as a distribution over token sequences, and in-distribution generalization represents the ability to successfully generate new samples from this distribution, while out-of-distribution generalization represents the ability to generate samples from a new distribution, given a few examples as context. In a nonlinear system, training on the pre-collapse attractor might cause a transformer to overindex on this specific distribution. Recent works showing that Transformers often approximate Markov chains on language datasets support the idea that these models could be treating dynamical attractors as fixed token distributions, leading to poor generalization.

**Metrics.** I like the idea of probing trained models for their implicit critical points. Bifurcation theory represents a natural way to formalize what “large” versus “small” distribution shift looks like, and the authors’ experiments can, in principle, be applied to any forecast model used to model a dynamical system.

The authors make an effort to fairly compare models. They try several variants of the transformer architecture, and they make sure to give the transformer as context the same information that they give to the RC as warmup, thus ensuring that both models see the same information.

**Exposition.** The quality of the motivation, exposition, and presentation is high. While I can’t give the paper a full endorsement at this stage, the idea and potential quality are both there, and I can see this paper improving substantially with revision. The paper is timely, particularly given recent interest in non-transformer models (like SSMs) that show signs of exhibiting comparable capabilities.

The Ikeda map demonstration is a great example, because, as the authors highlight, it contains a nested non-polynomial nonlinearity that cannot easily be expressed using standard terms in an equation library, thus establishing that that RC is doing more than just approximating the right hand side of the dynamics.

**Weaknesses:**

**Data leakage.** I am concerned that the use of parameter-aware RC makes the comparison unfair. While Wp in Eq. 10 is not trained, it provides a generalization signal because the authors train using three different values of the critical parameter, and so, implicitly, information about the directional effect of the parameter is available to the RC.
The authors also pass this parameter information to the transformer by appending a channel for it. However, it is not clear from the paper how exactly this is done (the transformer model is only briefly described in the appendix). Do the authors simply append a constant channel to the time series passed to the transformer during training?  If that’s the case, then wouldn’t the fairest comparison be to use a standard RC with that channel also appended to the input time series, rather than isolating it into a special input in Eq. 10? Or, if we want to use parameter-aware RC, we should instead benchmark against a conditional generative transformer.

**Novelty.** I’m a bit concerned that the results simply represent the bias-variance tradeoff in action. The authors repeatedly point out that the RC models use fewer parameters, and require less computing, in order to achieve a given accuracy. But, presumably, RC have substantial limitations in the classes of systems to which they are applicable, which is why they have so far proven unsuccessful for many general time series forecasting benchmarks. I understand that the typical argument in favor of RC is that they are better specifically for dynamical systems, but presumably, given a system I want to forecast, I rarely know in advance how much it acts like a deterministic system versus a (potentially smoothed) noisy one. Finding that a model with strong inductive biases outperforms a model with low inductive biases, particularly in the low-data or low-compute limit, is a standard expectation of the bias-variance tradeoff.

Along the same lines, this paper’s critique of transformers fails to engage with their broader capabilities that emerge as they scale, like in-context learning. Out-of-distribution generalization has been repeatedly shown for transformers on language tasks. Why aren’t the authors seeing it here? Is the claim that time series are somehow “special” relative to language? My concern is that the reason the authors aren’t seeing any generalization is the experiment design, or hyperparameter choices rather than a fundamental capability in limitation of transformers.

**Reproducibility.** The authors have not opted to make their code available for review. While this is not required by the conference, a study making strong normative claims about the relative merits of two methods should at least make the experiment design and setting clearer. This particularly concerns me, because the authors are benchmarking against a domain-specific model (RC) rather than standard choices like recurrent neural networks. How do we know that hyperparameters were chosen fairly? Were the transformers sufficiently regularized?

Overall, while I am sympathetic to the authors’ efforts to identify limitations of Transformers, the limited choice of datasets, limited baselines, issues with experiment design, and lack of reproducibility make the current paper’s claims overstated.

**Questions:**

1. Why not randomly vary the subcritical parameter across the dataset? Why pick exactly three values for each system? Most in-context learning experiments with transformers treat variation of in-context examples as continuous variables.

2. Can the authors confirm my understanding: the Transformers are given, during test, the exact same time series snippet used to warm up the reservoir computers? So the two models have access to equal information about the out-of-distribution case before they make predictions?

3. Can you clarify how the critical parameter information was passed to the Transformer?

---

> ### Author Response · Authors · 2025-11-21
>
> The informative and insightful comments provided by the referee are appreciated. The strengths highlighted by the referee, further clarify the motivation of our work. We have incorporated them into the revised manuscript (fourth paragraph of the Conclusion and Future Directions, page 9). Below we address the specific concerns.
>
> 1. For both RC and transformers, we use the same $(d+1)$-dimensional input: the $d$-dimensional state concatenated with a parameter channel. In RC, Eq. (11) [previously Eq. (10)] can be written as
> $W_{\text{in}} X(t) + W_p p = [W_{\text{in}}\, W_p]\,[X(t);\, p]$,
> by removing the parameter bias $p_0$, which shows that the parameter is simply appended as the last channel. We separate $W_{\text{in}}$ and $W_p$ only because they are drawn from different ranges to match the scaling of states and parameter. In the transformer, the parameter channel is appended in exactly the same way, and its input weights are adjusted during training. We clarify this in Appendix B.2, line 835-844.
>
> To ensure fairness, we also tested adding the same parameter bias term $p_0$ to the transformer during prediction for all three systems. As shown in Table 1, this modification does not affect the qualitative outcome: the transformer still fails to predict collapse.
>
> 2. We appreciate the referee’s insight on the connection to the bias-variance tradeoff. However, collapse prediction differs fundamentally from standard interpolation settings. The challenge is to extrapolate across a bifurcation diagram, where the underlying dynamical regime changes discontinuously and the attractor present in the training regime no longer exists in the test regime. In this setting, success depends not on variance reduction but on whether the model’s architecture encodes an inductive bias compatible with global phase-space geometry. RC succeeds because, as a dynamical system itself, it can form generalized synchronization with the target dynamics and track how trajectories change under parameter variations. Transformers, in contrast, rely on permutation-invariant attention and learn local sequence correlations without representing stability structure, invariant sets, or bifurcations. Even when we expanded the parameter range, increased model size, and performed extensive hyperparameter sweeps, all transformer variants we tested reproduced chaotic attractor instead of the post-collapse dynamics. In addition, regarding the NLP analogy, out-of-distribution generalization in language typically occurs under smooth distribution shifts. Here, the invariant measure changes non-smoothly at the critical transition, and the training data contain no collapsed-regime trajectories. For clarity, we provide a heuristic mechanistic discussion in Appendix G.
>
> 3. We agree with the referee on the importance of reproducibility. We have uploaded our code to OpenReview for review, and we will release the full repository publicly on GitHub. Different hyperparameters and transformer architectures are evaluated, as shown in Appendix F. In addition, We included additional transformer configurations, tests with pretrained foundation models, and an added spatiotemporal benchmark, the Kuramoto-Sivashinsky equation for comprehensive presentation.
>
> Q1. We chose three parameter values because this is the empirical minimum number required for RC to achieve stable and accurate performance. In addition to this, we also trained transformers on substantially richer parameter sweeps (up to 30 parameter values). The results, detailed in Appendix F (see Fig. 10 for an example), illustrate that the transformer still fails to predict system collapse.
>
> Q2. Yes, in the testing phase, both RC and transformers are given the data of length $L_{\max}$, which is the maximum sequence length in Transformer, for warm up.
>
> Q3. This point is addressed in our response to Comment 1.

---

> > ### Comment · Reviewer_L6LU · 2025-11-22
> > **Thank you for your reply, I will plan to review your comments this week**
> >
> > I thank the authors for their rebuttal and for making their code available. I will plan to review your rebuttal and revisions this week.

---

### Official Review · Reviewer_27Cm · 2025-10-28

**Soundness:** 2
**Presentation:** 2
**Contribution:** 2
**Rating:** 2
**Confidence:** 3

**Summary:**

This paper investigates whether Transformers can serve as faithful digital twins for nonlinear dynamical systems. Through experiments on few benchmark chaotic systems, it finds that while vanilla Transformers perform well in short-term forecasting, they may fail to predict critical transitions and collapses, unlike reservoir computing models.

**Strengths:**

The paper reveals a failure mode of vanilla Transformers in modeling parameter-dependent dynamical systems, providing a valuable benchmark and case into the limitations of current vanilla transformer models for physical dynamics.

**Weaknesses:**

## **Weaknesses**

1. **Overly broad and potentially misleading title**
   The title *“Can Transformers Truly Understand Dynamical Systems?”* is somewhat exaggerated and misleading.
   The paper only investigates a narrow empirical phenomenon — the failure of vanilla Transformers to predict bifurcation-induced collapses in a few low-dimensional chaotic systems — rather than addressing the full scope of “understanding dynamical systems.”
   A more precise title would better reflect the actual content and contribution.

2. **Unjustified generalization from vanilla Transformer to all Transformer architectures**
   The authors evaluate only a **vanilla decoder-only Transformer** and then generalize their conclusions to the entire Transformer family.
   This is not rigorous, as many recent Transformer variants — such as physics-informed, causal, continuous-time, or state-space Transformers — are specifically designed to handle temporal causality and dynamical structure.
   The observed failure may therefore stem from the limitations of the vanilla configuration, not from fundamental flaws in the Transformer paradigm itself.

3. **Limited experimental scope and absence of pretrained or large-scale models**
   The study trains small Transformers from scratch on a few parameter regimes.
   To make the conclusions more convincing, the authors should test **transformer-based pretrained or foundation models for dynamical systems**, such as *PANDA: A Pretrained Forecast Model for Chaotic Dynamics*, or other time-series foundation models (e.g., Chronos, TimesFM).
   Without such experiments, it is unclear whether the reported failure reflects model design or simply insufficient training diversity.

4. **Lack of theoretical insight beyond empirical observation**
   While the paper aims to argue that Transformers fail to learn the true underlying dynamics, the evidence is entirely empirical.
   The superior performance of reservoir computing can already be explained by its established theoretical grounding in dynamical systems (e.g., generalized synchronization, echo state property).
   In contrast, the Transformer’s failure is not analyzed from a mathematical or dynamical perspective.
   Without deeper theoretical or mechanistic insight — such as linking self-attention to Lyapunov stability or bifurcation sensitivity, the work remains observational rather than explanatory.

5. **No code release for reproducibility**
   The paper does not mention any code or data release.

**Questions:**

## **Questions for Authors**

1. **Model generalization**
   Why do the authors generalize their findings from a vanilla Transformer to all Transformer architectures?

2. **Experimental scope**
   Have the authors considered testing pretrained or large-scale time-series models such as *PANDA* or *Chronos*?

3. **Lack of theoretical insight**
   Can the authors provide any theoretical analysis or dynamical explanation for why Transformers fail to capture bifurcation behavior?

---

> ### Author Response · Authors · 2025-11-21
>
> We thank the referee for the thoughtful and constructive comments. We have revised the manuscript accordingly, within our computational and time constraints.
>
> 1. We agree with the referee that the title should be more precise. We have changed it to ``Can transformers predict system collapse in dynamical systems?''
>
> 2. Our study focuses on the vanilla transformer, the canonical and most widely deployed architecture. The transformer family has grown extremely large, with hundreds of variants. It is practically impossible to evaluate all of them systematically. In our manuscript, we state that our findings are based on the vanilla transformer across different configurations, under a broad range of hyperparameters and data sizes, and parameter sweeps are tested. Transformers persistently fail on collapse prediction.
>
> The aim of this study is to report a surprising and well-controlled empirical contrast. That is, transformers perform well in the safe regime, but consistently fail to predict system collapse. In comparison, RC performs well in both safe and collapse regimes, with significantly less data and tuning. Nevertheless, we agree that evaluating more specialized variants is an important direction. We have added the statement to Conclusion and Future Directions, in line 490-493.
>
> 3. We appreciate the referee’s suggestion and therefore tested two representative foundation models, PANDA and Chronos-2. PANDA is a milestone pretrained model for chaotic dynamical systems. In the safe regime, it provides accurate short-term forecasts but its long-term predictions deteriorate and drift toward a fixed point [e.g., Fig. 8(a)]. Because reliable long-term forecasting is a prerequisite for collapse detection, PANDA does not qualify for our task. Nevertheless, we evaluated it under collapse conditions and found that it again predicts short-term accurately but still converges to an incorrect fixed point, failing to detect collapse [Fig. 8(b)]. Chronos-2, a multivariate extension (Chronos and TimesFM are restricted to univariate inputs), performs reasonably in short-term predictions but, like PANDA, cannot reconstruct chaotic attractors in long-term forecasts and therefore also fails to detect collapse. These results are summarized in Appendix E.
>
> In addition, all of our experiments in foundation models use zero-shot inference. Fine-tuning large foundation models is nontrivial and computationally expensive. We have noted this in the Conclusion and Future Directions, line 489-490.
>
> 4. We completely agree with the referee that the theoretical or mechanistic insights are important. We have added a new section, Appendix G to heuristically compare RCs and Transformers.
>
> 5. We have uploaded our codes through OpenReview, as Supplementary Materials for anonymous review. After review, we will release the full repository publicly on GitHub.
>
> The questions raised by the referee overlap with the points discussed above, and are addressed in our response.

---

### Official Review · Reviewer_iggh · 2025-10-30

**Soundness:** 4
**Presentation:** 3
**Contribution:** 3
**Rating:** 8
**Confidence:** 4

**Summary:**

The authors challenge the efficacy of transformers for learning from dynamical systems, and then begin able to produce accurate predictions in unseen regimes during training. In particular, the authors challenge transformers against being able to predict the phenomenon of “system collapse” in chaotic systems. They choose 3 low-dimensional parametrized chaotic dynamical systems, which exhibit system collapse only for after some critical parameter value. They generate a training dataset by solving these systems at values below the critical parameter, and they include these parameter values as additional information in the training dataset. This results in no information about the system collapse phenomenon being passed to the model during training. After making sure the model is trained properly on the parameter range it has seen in its training dataset, it is tested beyond the critical parameter point where collapse is expected, and it is evaluated by checking whether it is able to predict the collapse phenomenon. This experimental methodology is applied to 2 types of models: the transformer which is the model under investigation, and to Reservoir Computing (RC) models as a benchmark to compare against. What the results show is that the transformer fails catastrophically at deducing the collapse phenomenon, which highlights limitations that Transformers have in their current state. But also, the alternative model tested by the authors (RC) shows remarkably good ability to capture the collapse phenomenon, a very impressive result.

**Strengths:**

- The results of the paper are strong, the transformer architecture catastrophically fails at the prediction task designed, while the success of the proposed alternative proves that the task is not unreasonably difficult. Further, failure of the transformer architecture is justified in the sense that the training setup is fair and unbiased (see more below)
- A novel methodology for evaluating the ability to extrapolate to unseen phenomena in the training dataset is established. This sets a significant precedence beyond the types of architecture relevant to this paper (RCs and Transformers).
- As mentioned, the authors do not only address the stated research question but also provide a strong alternative (RC) which is very successful in the difficult prediction task designed.
- The investigation of the research question is thorough and detailed. Many potentials pitfalls to the experimentation methodology are addressed such as the fact that there may be a model specific critical parameter beyond which the model collapses, or the fact that the model may eventually collapse if the trajectories are propagated further in time. The reader feels confident on the experimentation (see ‘Weaknesses’ for more details) and the choices taken by the authors are justified.
- The presentation of the material is of high quality, providing plenty of clear discussion and interesting avenues for future research, such as the attempt to mitigate the scaling limitations of RC by defining a hybrid approach combining them with transformers. Perhaps most interesting is the attempt to better understand the success of RCs (and respectively failure of transformers), in order to design better methodologies for learning from dynamical systems.

**Weaknesses:**

- The claim that (Line 88-89) “…this is the first work to systematically challenge the effectiveness of transformers as digital twins of dynamical systems” seems a bit too broad in scope.  A potential counterexample could be the 2023 paper published in IEEE CSS with title “Can Transformers Learn Optimal Filtering for Unknown Systems?”, Du Z. et. al. Limiting the scope of the statement, by e.g., specifying the chaotic phenomenon under investigation (system collapse), or even just the fact that this paper’s focus is chaotic dynamical systems, will improve on this point.
- Another point of criticism is the complexity of the 3 example dynamical systems chosen. The authors do mention that the last one (Ikeda map) is a particularly challenging one, and the examples are clearly “difficult-enough” to show that the Transformer architecture fails in the task. Nevertheless, the RC seems to be highly performant in all of them. In this respect, it could be beneficial (in order to highlight the limitations of the alternative proposed) to have included an example where the RC is showing significantly lower performance, perhaps something of higher dimensionality, since as the authors say one of the limitations of RC is its scalability issues. In particular, in a higher dimensional chaotic system exhibiting system collapse (even an artificially constructed one, made by building on one of the examples in a cartesian way), these scalability issues of RC might become more apparent. In any case, RC is not the main thesis of this paper, so I consider this a minor point, especially when weighted against the difficulty of incorporating an extra example.
- A final minor point is the convergence of the reported statistics. It could have been good to include in the appendix (which is not being reviewed) some indicative convergence plot for the statistics being reported in Table 1. There is not really a serious doubt about convergence, though a convergence plot with respect to number of simulations (in this case 1000) would have increased the confidence of the reader even further.

Minor Typos/ Styling **Suggestions**:
- Line 51-52: “An interesting perspective for interpreting RCs is that the dynamics…”
- Line 52: Replace “an generally” with “a generally”
- Line 107: Consider removing “for dynamical systems”.
- Line 203: Expand “NLP” definition, it is not defined anywhere above.
- Line 318-319: I’m having trouble interpreting the last sentence, it has some grammatical errors. Consider changing it to “In this setting, none of our experiments with Transformers resulted in success, despite extensive tuning…”.
- Line 401-402: Consider removing “but then”.

**Questions:**

Questions:

- When it comes to deducing the collapse phenomenon not observed in the training dataset, I am curious whether it is fair to expect this from any model. The answer is seemingly positive, since RCs perform so well, but still I wonder if it is possible to design a counterexample by adjusting one of the presented dynamical systems, such that its restriction on the training parameter range performs **identically** with the original one, but beyond the critical parameter it does not exhibit collapse, but remains oscillatory in nature. If such an example exists, then the results would be reversed, RCs catastrophically failing and Transformers succeeding (probably). I would like to better understand whether such an example is possible in 3D (resp. 4D or Complex1D for the other examples) or if there are smooth continuation constraints that do not allow it to exist. Do you think that such an example can be designed? If not, why? If yes, how come the RCs predict accurately?
- I noticed in Figure 3d, that the RC collapse is different than the system's collapse (they collapse to different values). Granted, the collapse phenomenon is predicted by the RC, but I am wondering if you have any suggestions for designing a methodology for RCs that would allow not only the prediction of collapse phenomena, but also the exact way the collapse will take place.
- Have you/ did you consider other phenomena than system collapse? I am curious what would be the next phenomenon you would test against, perhaps more challenging than system collapse, where you expect RCs to begin having trouble. In general, (though I definitely am not suggesting you attempt to include this in the paper, given how much effort is required) I would be very curious to see a table similar to Table 1, where an array of different phenomena are evaluated.

---

> ### Author Response · Authors · 2025-11-21
>
> We appreciate the referee's positive evaluation and recommendation, especially by staring ``the results of the paper are strong, the presentation is of high quality, providing plenty of clear discussion and interesting avenues for future research.''
>
> 1. We have revised the statement in line 88-90 into ``this is the first work to systematically challenge the effectiveness of transformers as digital twins in dynamical systems for anticipating system collapse,'' and cited related works, including the reference pointed out by the referee (line 126-128).
>
> 2. We appreciate the suggestion to include a higher-dimensional system. We have added a paradigmatic model of spatiotemporal chaotic systems, the Kuramoto-Sivashinsky (KS) equation. RC performance somewhat degrades in this more complex setting (representative results shown in Fig. 6 with statistical summaries in Table 2, Appendix). Statistically, the $P_c$ (the critical point of collapse) is 0.94. However, the transformer continues to perform poorly in detecting the collapse. This is reasonable, as it has already been demonstrated that transformer fails on low-dimensional systems, and there is no reason to expect that it would succeed on more complex, higher-dimensional chaotic systems. Details are provided in Appendix A.3 and Appendix C.3.
>
> 3. We have corrected the typos as suggested, and rechecked our manuscript for better presentation.
>
> Q1. This is an insightful question. There are works show that RCs can reconstruct entire bifurcation diagrams, including transitions between periodicity and chaos, by training on several distinct bifurcation parameter values (e.g., Kong et al., 2023). This explains why RCs can extrapolate beyond the training parameter regime in our experiments: it learns the underlying relationship between parameter and dynamics, rather than just the trajectories. In addition, a systematic study of whether transformers can similarly reconstruct bifurcation diagrams within the safe regime is an interesting topic for future research.
>
> Q2. This is an important but largely open problem. Predicting where the collapse settles requires accurately capturing the global structure of the attractor basin beyond the critical transition. This is a notoriously difficult problem in different fields of science and engineering. While RCs can reliably detect the onset of collapse, predicting the post-collapse equilibrium remains challenging, as no training data would be available in the post-critical regime. Injecting additional physical knowledge or structural constraints may improve this.
>
> Q3. System collapse is arguably the most abrupt and challenging phenomenon in dynamical systems. While RC performs reasonably well, especially for low-dimensional systems, transformers fail, indicating the difficulty of predicting sudden transitions. Beyond that, other benchmarks include safe regime reconstruction and performance on limited data.
>
> We have stated these points as the third paragraph in Conclusion and Future Directions, on page 9.

---

### Author Response · Authors · 2025-11-21
**Summary of main changes**

In response to the referees' comments, we have

1. revised our title to ``Can transformers predict system collapse in dynamical systems?''

2. clarified heuristically the differences in the main mechanism between reservoir computing and transformer,

3. provided new results on the spatiotemporal Kuramoto-Sivashinsky system,

4. evaluated different transformer configurations and foundations models on system collapse prediction,

5. updated our codes on OpenReview,

6. corrected minor issues, such as typos, symbols, and grammars.

---

### Author Response · Authors · 2025-11-21
**General reply**

Dear Area Chair:

Thank you very much for obtaining four referee reports on our manuscript #9664 previously entitled ''Can transformers truly understand dynamical systems?'' (revised title: ``Can transformers predict system collapse in dynamical systems?''). We have carefully revised the paper to fully address all the referee comments. A summary of the main changes made and a point-to-point response to all referee comments are provided. The changes in the text are marked blue.

We believe that a thorough discussion with all referees would help further clarify any remaining concerns, so that they can choose to reassess our paper. We sincerely appreciate your assistance throughout this process.

We would like to thank you again for your wonderful and professional handling of our manuscript. We hope our revised manuscript can be judged to have met the high standards of ICLR.

Best regards,

All authors

---

### Author Response · Authors · 2025-11-21
**General reply**

Dear Referees:

We thank you all for your careful evaluations and thoughtful feedbacks on our manuscript. Your comments have been extremely valuable, helping us better understand how researchers from different perspectives view this line of work, and inspiring several directions we had not previously considered.

A point-by-point response to each report has been uploaded, and all changes in the text are mark in blue. We greatly appreciate the time and effort you invested in the review, and we hope that our revisions adequately reflect your suggestions.

We welcome any further feedback you may have on the updated version, and we would be glad to continue constructive discussion as needed. Thank you very much for your time and consideration.

Best regards,

All authors

---

### Meta-Review · Area_Chair_ryLn · 2026-01-08

**Summary:**

The paper studies whether transformer predicts the collapse in dynamical systems (compared to reservoir computing (RC)). Interesting enough it is found that transformer models can not extrapolate well in this scenario.
Main reviewer concerns:
1) The original title was too broad (about 'understanding'), the authors changed it;
2) The comparison to RC may be subject to data leakage
3) Not enough experiments for larger models/other architectures/pretrained models.
4) No code released for reproducability
5) No theoretical justification.

**Reviewer Concerns:**

Reviewer iggh: put the high score (8), addressed (the reference to a 2023 paper)
Reviewer 27Cm (score 2), but most of the comments are general and addressed (more experiments, the code, the title), but the theoretical part is not.
Reviewer L6LU (score 4): concern about bias-variance tradeoff (RC has fewer parameters)  - answered vaguely in the text
Reviewer aMYQ (score 4):  concern about no theoretical understanding.

I think the concern by answering the question 'why' transformer is not generalizing in this scenario is not addressed or verified.

**Reviewer Scores:**

iggh: 8
27Cm: 2 -> 4
L6LU: 4 -> 6
aMYQ: 4

---

### Decision · Program_Chairs · 2026-01-26

Reject